# Read my LIPSS: organic lasers on micromachined resonators

Tiange Dong [1,4] ✉, Tobias Antrack [1,4] ✉, Frithjof Pietsch[1], Jakob Lindenthal [1], Markus Löffler[2], Bernd Rellinghaus [2], Johannes Benduhn [1,3], Markas Sudzius[1] & Karl Leo[1] ✉

Thin-film lasers based on organic semiconductors offer significant potential for miniaturized sensing and optical memory. We present a new method for creating first-order distributed feedback (DFB) Bragg gratings using laser-induced periodic surface structures (LIPSS) as the optical resonator. These subwavelength structures, fabricated via femtosecond laser micromachining, exhibit stable periodicities and provide sufficient feedback for lasing in an $Alq_3$:DCM film. The laser emission wavelength can be widely tuned by varying the LIPSS periodicity defined by the pulse spacing of the structuring laser. While lasing is limited by LIPSS imperfections, we demonstrate that individual LIPSS gratings optically couple into a coherent macroscopic supermode. This coupling dramatically increases lasing efficiency and reduces thresholds by two orders of magnitude compared to a single LIPSS element. This straightforward fabrication enables solid-state organic DFB lasers as integrated, on-chip coherent light sources, eliminating complex external coupling into photonic circuits.

Organic semiconductors are an excellent material choice for the cost-effective production of light-emitting devices[1,2]. Their spectrally broad absorption and emission spectra in the visible range, high optical gain, and simple processing make them particularly interesting for lasing applications[3,4]. Recently, considerable efforts have been made to optimize materials and design towards higher efficiency, lower lasing threshold, and lower cost[3,5,6]. In general, simpler and faster fabrication methods consequently result in lower production costs.

All laser devices require some form of positive optical feedback. In organic solid-state microresonator systems, this is typically achieved via a Fabry-Pérot type cavity or distributed feedback (DFB) structures. While Pérot cavities are perhaps the more intuitive cavity arrangement[7], they consist of numerous layers, which results in a bulky design that requires significant time and hence cost to be fabricated. DFBs, on the other hand, offer a more compact design where the resonator structure is integrated within the gain medium. These structures combine inherent waveguide confinement with periodic in-plane perturbations, resulting in improved optical confinement and modal gain[3]. This can lead to lower lasing thresholds compared to more complex resonator structures[4,6,8–10]. While lower Bragg orders in DFB lasers offer clear advantages in terms of lasing threshold, realizing these gratings for the visible light spectrum requires nanopatterned substrates with characteristic periodicities in the range of a few hundred nanometers[11].

The most common techniques for fabricating subwavelength gratings include electron-beam lithography[8,12–15], interference lithography[16–18], and photolithography[8,19–22]. These methods are typically followed by chemical etching, resulting in multiple processing steps and imposing strict requirements for the underlying materials. As a one-step process, nanoimprint lithography is often used[11,23]. However, other lithography processes are still required to create new molds. Recently, direct laser writing was applied as a fast and flexible

[1]Dresden Integrated Center for Applied Physics and Photonic Materials (IAPP) and Institute of Applied Physics, Technische Universität Dresden, Dresden, Germany. [2]Dresden Center for Nanoanalysis (DCN), Center for Advancing Electronics Dresden (CFAED), Technische Universität Dresden, Dresden, Germany. [3]Deutsches Zentrum für Astrophysik, 1, Postplatz, 02826 Saxony Görlitz, Germany. [4]These authors contributed equally: Tiange Dong, Tobias Antrack. ✉e-mail: tiange.dong@mpi-halle.mpg.de; tobias.antrack@tu-dresden.de; karl.leo@tu-dresden.de

technique to create first-order gratings in waveguides[24–28] for applications in the infrared spectrum region. However, in all of these approaches, the smallest achievable feature size is constrained by the diffraction limit[29], making direct laser structuring an unsuitable candidate for the fabrication of first-order gratings in the visible light spectrum.

Alternatively, applying multiple pulses from a linear polarized pulsed laser with power densities below the single-shot ablation threshold can lead to the formation of laser induced periodic surface structures (LIPSS)[30]. These surface ripples can be produced on almost all types of surfaces[31] and can be divided into two classes: low-spatial-frequency LIPSS (LSFL) with spatial periods close to the utilized laser wavelength and orientation parallel to the electric field vector for large band gap materials[32], and high-spatial-frequency LIPSS (HSFL). With pulse lengths in the femtosecond range, HSFL can form on most transparent materials. HSFL is characterized by spatial periods significantly smaller than the irradiation wavelength and orientations perpendicular to the polarization for most materials[33]. However, the exact microscopic processes involved in the formation of HSFL are still under research, with possible explanations including material self-organization and multi-photon absorption[32–37].

In this work, we combine the advantages of organic DFB lasers with the flexibility and simplicity of LIPSS fabrication. This approach is a promising method for fabricating organic thin film structures with tunable lasing properties. These lasers use thermally deposited thin films of small organic molecules as the gain medium on a periodically patterned substrate. The patterning is achieved by femtosecond laser micromachining, providing unique flexibility in fabrication speed and wave pattern characteristics suitable for DFB lasing. Using our presented one-step direct laser writing technique allows for the rapid fabrication of a wide variety of model photonic systems, enabling a comprehensive study of their optical and lasing properties. Our experimental results demonstrate that the periodicity and orientation of LIPSS can be precisely controlled, providing sufficient optical feedback from the corrugation to achieve lasing in first-order DFBs. We also show that systematically distributed LIPSS enable coherent optical coupling of adjacent structures, forming single large-area photonic lattices. These artificially designed periodic photonic structures, intentionally inscribed to support optical supermodes, effectively remove the inherent constraints on lasing mode volume size. As a result, lower lasing thresholds are achieved.

## Results and discussion

### Femtosecond laser fabrication of LIPSS gratings

To fabricate a corrugated surface that serves as a DFB reflector, we employ femtosecond laser illumination to precisely modify the substrate surface. This process involves inscribing LIPSS into glass substrates through line scanning with a pulsed laser (515 nm wavelength, 200 fs pulse duration, 50 kHz pulse repetition rate), as depicted in Fig. 1a. By carefully choosing the pulse fluence, the distance between adjacent laser shots, and the scanning speed, we are able to achieve a structuring regime that is favorable for LIPSS formation. With the structures having periodicities significantly below the structuring laser wavelength, and polarization being parallel to the scanning direction (and consequently perpendicular to the resulting LIPSS ripples, see Supplementary Fig. 1), we are in the regime of high-spatial frequency LIPSS (HSFL) for glass as the substrate[30,32]. The effective repetition rate of structuring pulses impinging on the sample was varied around 5 kHz. Combined with a scanning speed of 1 mm/s, this resulted in a shot-to-shot spacing ranging from 183 to 195 nm. Under these conditions, increasing the pulse-to-pulse spacing also leads to an increased LIPSS periodicity $\Lambda$. Notably, in contrast to previous works[38–40], we observe a linear relation between the pulse spacing and the resulting LIPSS periodicity, where the pulse spacing equals $\Lambda$ (see Fig. 1b). Moreover, we did not notice a dependency of $\Lambda$ on the laser fluence[41] (see Supplementary Fig. 2). Both observations indicate

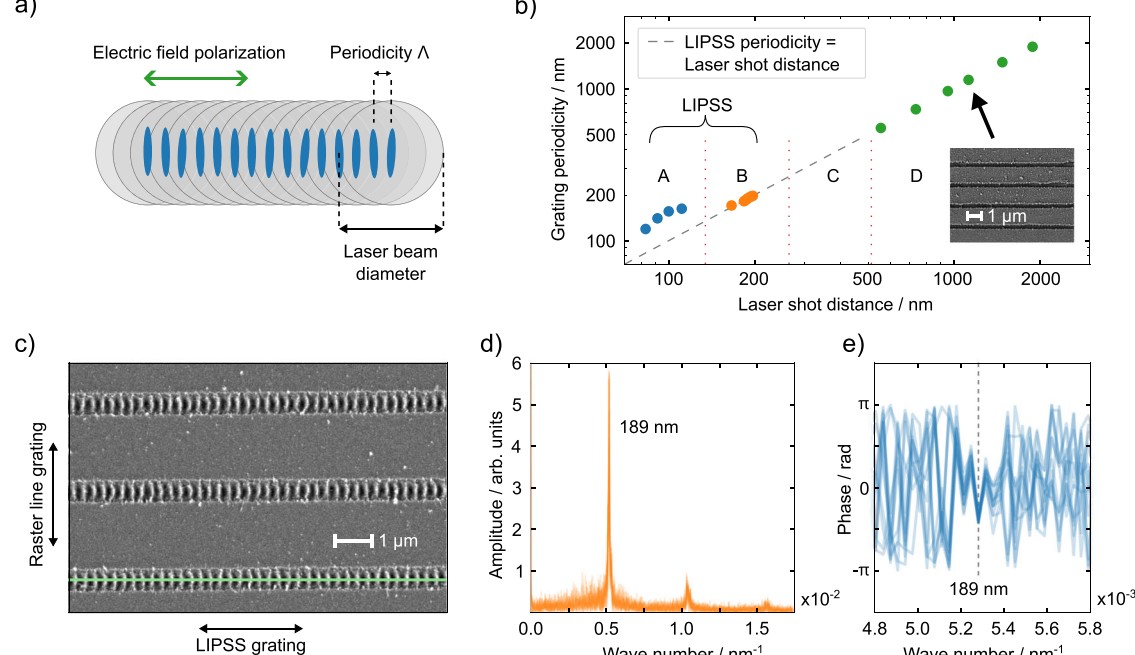

**Fig. 1 | Fabrication of LIPSS. a** Schematic of femtosecond laser scanning to produce laser-induced periodic surface structures (LIPSS), where the direction of the resulting LIPSS grating lines is perpendicular to the polarization of the incident laser beam. **b** Dependency of resulting LIPSS periodicity on the lateral structuring laser pulse distance. The resulting LIPSS periodicity is larger than the laser pulse spacing in regime A. In regime B, the periodicity equals the pulse spacing, whereas in regime C, no grating can be created. Regime D denotes gratings produced by ablation (writing grating lines directly with the laser). **c** Exemplary top view SEM image with the green line marking the pixels analyzed for periodicity and phase of the LIPSS gratings. **d** Amplitudes of the Fourier spectrum of 13 analyzed LIPSS gratings with a lateral distance of 1.47 μm and **e** the corresponding Fourier phases. See Supplementary Figs. 5, 8 for more details.

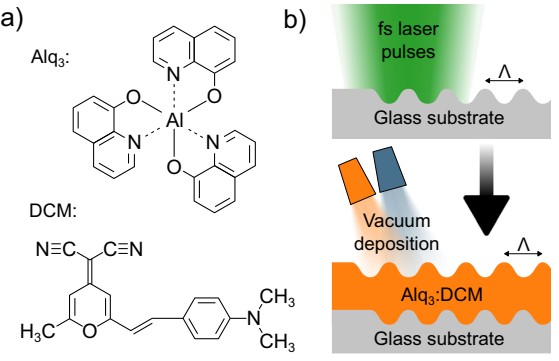

**Fig. 2 | Deposition of organic gain material. a** Chemical structures of Alq$_3$ and DCM. **b** Schematic representation of sample fabrication: LIPSS fabrication with focused pulsed laser beam followed by thermal evaporation of 500 nm Alq$_3$:DCM on top of the structures. **c** Cross-sectional SEM image of a LIPSS grating with the deposited organic layer on top. The gold and platinum layers are deposited as conductive and protection layers for the cutting process and SEM observation.

that rim-formation[42] due to single point writing[39] is also involved in the grating formation (see Supplementary Note 1 for extended analysis). The depth of the surface ripples increases with the laser fluence from around 30 up to 40 nm (see Supplementary Fig. 3), and the width of the LIPSS stripe varies from 130 to 400 nm (see Supplementary Figs. 2, 9). With a laser focus point size of 1.1 μm and a pulse fluence of around 80 nJ, the energy density during fabrication is comparable to previously reported threshold energy to fabricate LIPSS[43].

Fast Fourier Transformation (FFT) of a top-view Scanning Electron Microscopy (SEM) image (Fig. 1c) reveals that neighboring LIPSS lines have the exact same carrier frequency (Fig. 1d). Moreover, individual LIPSS lines have the same phase for this frequency, which is locked (Fig. 1e). This allows effective optical coupling between adjacent LIPSS lines and establishes coherence onset throughout the whole structure. As we will show later, this coupling has a significant effect on the lasing characteristics of the system.

The FFT amplitude spectrum also shows second and third peaks corresponding to higher orders of the carrier frequency. These additional peaks indicate the presence of higher harmonics and suggest deviations from a purely sinusoidal spatial profile. It is important to note that the presence of these higher harmonics can be attributed to several factors, including the inherently complex material self-assembly process that occurs during LIPSS formation under femtosecond multi-shot laser treatment[44]. This anharmonicity in the spatial profile highlights the sophisticated nature of the LIPSS formation mechanism and its potential impact on the optical properties of the resulting structure.

## LIPSS Bragg gratings

DFB lasers are based on positive optical feedback created by periodic structures within a waveguiding layer where the resonant wavelength is defined by the Bragg condition: $\lambda = 2\Lambda n_{\text{eff}}/m$. Here, $\Lambda$ is the grating periodicity, $n_{\text{eff}}$ the effective refractive index, and $m$ the Bragg order. Using the first Bragg order generally gives stronger optical feedback than higher orders[45]. With LIPSS, we create highly periodic ripples on the substrate surface, having periodicities sufficiently small to act as a first-order Bragg grating within the visible light spectrum. Our organic layer simultaneously acts as both the waveguide and the emissive layer. It has a thickness of 500 nm, which is sufficient to ensure proper waveguiding of spontaneously emitted light and support lasing in the visible spectrum.

Tris-(8-hydroxyquinoline) aluminum (Alq$_3$) and 4-(dicyano-methylene)−2-methyl-6(p-dimethylaminostyryl)−4H-pyran (DCM) are co-evaporated to create a 2 wt% doped emissive layer (Fig. 2a, b). The crucibles are located with a slight offset underneath the substrate inside the vacuum chamber during material deposition, which results in the molecules approaching the substrate under an angle of around

5°. In Fig. 2c, a cross-section of a LIPSS line with the organic layer on top is presented, where it is clearly visible that the surface modification of LIPSS is transferred to the top surface of the organic layer. However, the aforementioned position of the crucible leads to a shift of the wave-like surface shape with increasing layer thickness. In our case, the final thickness of the organic layer combined with the deposition angle leads to a phase difference of roughly π between the bottom and top layer modification. As a result, the entire organic layer experiences a thickness variation along the grating vector of the underlying LIPSS structure.

## Lasing under optical pumping

For optical characterization, the fabricated first-order organic DFB lasers are optically excited using femtosecond laser pulses with a wavelength of 400 nm. Lasing is identified by the emergence of a sharp peak at the top of the photoluminescence spectrum as the pump fluence is increased, as illustrated in Fig. 3a. Additionally, the characteristic non-linear increase in emission intensity, accompanied by a sudden reduction in the full width at half-maximum (FWHM), is presented in Fig. 3b. The lasing threshold, defined by the onset of non-linearity in the input-output curve, is experimentally determined to be as low as 185 μJ/cm$^{-2}$. Compared to other works, which use the same materials but a DFB grating fabricated by electron-beam lithography[12], our threshold is more than one order of magnitude higher, primarily due to the inherently lower optical quality of gratings produced via direct laser structuring. When the shot-to-shot spacing during LIPSS fabrication is varied, the periodicity of LIPSS is varied. Consequently, the DFB resonance frequency changes, resulting in a shift of the lasing wavelength as shown in Fig. 3c. This is clear evidence that the LIPSS structure itself provides the positive feedback by fulfilling the first-order Bragg condition, which allows for lasing to begin. As further evidence, Fig. 3d shows that there is no lasing observed when an unstructured region is optically pumped.

By comparing the LIPSS periodicity $\Lambda$ with the measured lasing wavelength based on the Bragg condition, we obtain an effective refractive index of $n_{\text{eff}} = 1.64$ at 618 nm. This fits well to the calculated value for the waveguided mode (see Supplementary Fig. 10). Owing to the spectral resolution limit of our detection system (about 0.2 nm) and the large angular integration used during measurement, the recorded emission peaks are artificially broadened. As a result, this does not allow us to reliably estimate the spectral width of the lasing line (see Fig. 3b).

## Lasing threshold in variable length LIPSS DFB lasers

The ability to precisely control and easily scale the size of laser-inscribed photonic structures allows us to systematically investigate lasing characteristics as a function of resonator geometry. A larger

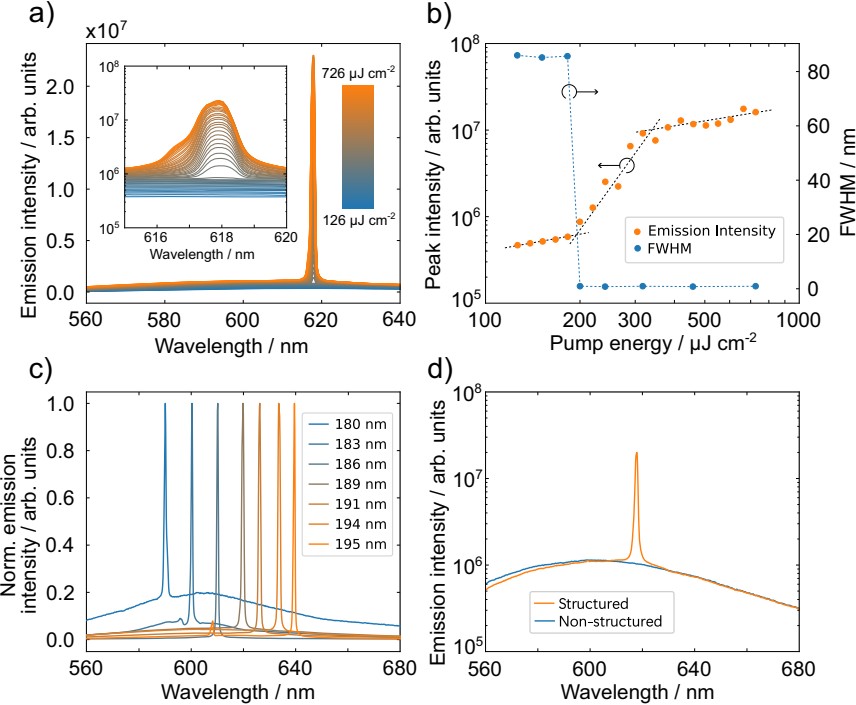

**Fig. 3 | Lasing behavior under optical pumping. a** Emitted spectra with increasing pumping intensity (blue to orange). The inset shows the lasing emission peak in detail and on a logarithmic y-scale. **b** Input-output characteristics derived from the analysis of signal peak intensity with clearly visible s-shaped curve and a drastic decrease of FWHM at a lasing threshold of 185 μJ/cm$^{-2}$. The dotted lines are a guide to the eye. **c** Emission spectra obtained from pumping LIPSS structures with different periodicities ($\Lambda$ = 180 nm … 195 nm). **d** Comparison of the emission spectra under the same pump fluence of an area with LIPSS structures and an area without any structures.

mode volume enables more molecules to participate in the lasing process, leading to enhanced output intensity. Simultaneously, the increased net gain associated with a larger mode volume reduces the lasing threshold. Exploiting the high versatility of our fabrication method, we prepared a series of samples with varying LIPSS stripe lengths to precisely control the DFB grating length, as shown in Fig. 4a. With this approach, we are altering one dimension of the lasing mode volume and aim to understand the influence of the cavity quality on the lasing behavior. As the stripe length decreases, the mode volume becomes constrained by the size of the inscribed feedback structure. When the stripe length falls below the effective mode volume, a pronounced increase in the lasing threshold is expected. Figure 4b shows the lasing threshold energy density as a function of the LIPSS grating length, where a clear decrease of lasing threshold for longer stripe lengths with subsequent saturation is visible.

The experimentally observed threshold behavior is in good agreement with the results of previous studies on the lasing threshold of optically pumped organic DFB lasers as a function of the pump beam spot size[46,47]. These studies showed that the threshold rapidly decreases with increasing pump spot size in a strongly nonlinear way until a critical pump spot size is reached, at which point the threshold levels off. This transition from the nonlinear to the saturation regime can be qualitatively explained by coupled wave theory[48], where the effective device length is related to the pump spot diameter and the coupling coefficient of the guided modes is primarily governed by the refractive index perturbation. The decrease of the threshold is explained by the gradual improvement of the coupling efficiency between the counterpropagating waves with increasing grating length, until the mode "overcoupling" regime is reached[47], or the lower lasing threshold regime is reached due to the gain material limitations. For a 2$^{nd}$-order DFB laser system based on high optical quality Bragg grating resonators and different organic gain materials, the critical grating length was found to be in the range from 200 to 350 μm[46,47]. In our

case, however, the critical grating length is around 10 μm and is more than one order of magnitude shorter than those reported above. We attribute this observation to the fact that DFB lasers heavily rely on phase perturbations of the resonator gratings and are therefore extremely sensitive to grating non-uniformities. As shown above, LIPSS gratings are prone to significant surface profile disturbances (see e.g., Figs. 4a, c and S5), which affect the grating periodicity and consequently its optical quality due to the formation of higher Fourier spatial harmonics and significant spectral noise onset. As a result, no DFB resonance can be observed in the linear regime below the lasing threshold.

Since analytical approaches such as coupled wave theory or a simple rate equation model cannot properly account for the random nature of the phenomena, we use a finite-difference time-domain (FDTD) method to simulate the optical response of our systems based on experimentally measured realistic LIPSS spatial profiles, as presented in Fig. 5a. Using numerical simulations, we show in Fig. 5b that in our system, the spatial profile inhomogeneities are the dominant factor limiting the coherence build-up along the LIPSS grating vector. Thus, they can severely limit the in-plane coherence extent of the light and consequently the lasing thresholds in LIPSS-based DFBs.

In Fig. 5c, d, the calculated reflectivity spectra of two corrugated optical waveguides are shown. One waveguide is based on an experimentally measured AFM substrate surface profile, while the other is based on an analogous, but perfectly periodic, grating with the same fundamental spatial frequency, similar duty cycle, and characteristic grating depth as the real grating. In these calculations, the light source is positioned inside the waveguide next to the grating, and the percentage of reflected power is simulated as a function of the periods used to form a grating for different wavelengths. All optical constants are chosen to mimic the real system as closely as possible (Table S1). Both systems demonstrate an effective build-up of the Bragg reflectivity band over just a few hundred periods, followed by saturation of

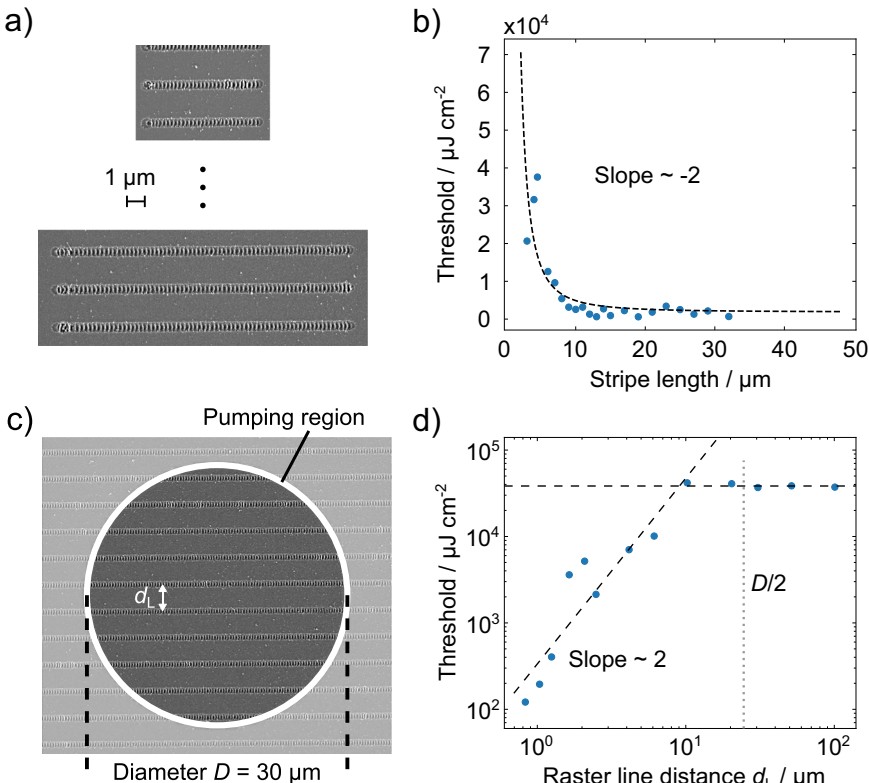

**Fig. 4 | Coupling behavior and effective mode volume. a** SEM images of LIPSS stripes with constant line separation but varying stipe length, and **b** the measured lasing thresholds for different stripe lengths. The threshold is decreasing non-linearly with increasing stripe length until saturation is reached. **c** SEM image of a LIPSS-structured glass substrate and exemplary representation of the area being excited by the pumping beam. **d** Lasing threshold as a function of the raster line distance $d_L$ between adjacent LIPPS lines for a fixed pump spot size of 30 µm. The threshold reaches a saturation just before $d_L$ is around half of the pump spot diameter $D$. For $d_L > D/2$ effectively only one LIPSS line is being excited. The dashed lines are linear fits of the measurement data.

reflectivity with its maximum at the center of the stopband. The spectral stopband position coincides perfectly with the experimentally observed relation between grating period and lasing wavelength. The reflectivity saturation levels sensitively indicate structural perturbations that may be present in Bragg gratings and highlight the effect on how efficiently they can manage feedback and coherence of light, thereby influencing the quality factor of the DFB resonator. Similar to the photoluminescence spectra below the lasing threshold, no resonances can be observed in the reflectance spectrum.

The peak reflectivities are explicitly compared in Fig. 5b. While the reflectivity of the perfect grating asymptotically approaches 100% as a function of the number of grating periods, the reflectivity of the grating based on the experimentally measured spatial profile begins to saturate at about 80% due to profile imperfections. In other words, 80% reflectivity in a perfect grating would be achieved with a grating length of approximately 150 periods. This comparison suggests that due to imperfections in the LIPSS grating profile, the effective length along the LIPSS grating vector that contributes to coherence build-up is approximately 28 µm. This is a rough estimate, as our calculation ignores the spatial dimension along the LIPSS grating lines, which is likely to further reduce the coherence length. Experimentally, as shown in Fig. 4b we obtain a critical grating length for the LIPSS DFB of about 10 µm, where the coherence build-up levels out and the lasing threshold saturates to its minimum value of 185 µJ/cm², characteristic of this LIPSS-based DFB system.

**Coherent optical coupling between LIPSS lines**
As discussed earlier, the periodicities of LIPSS gratings on glass produced by laser structuring can be far below the diffraction limit of the inscribing beam, while still exhibiting stable dominant spatial

frequencies and phases (see Fig. 1d, e). On the other hand, if the distance between adjacent lines is less than the coherence length of the spontaneously emitted light and small enough to allow efficient optical coupling, lateral coherence can begin to develop and spread throughout the size of the optical pump beam. Such coupling effectively increases the modal volume where the positive optical feedback is supplied, resulting in an efficient reduction of lasing thresholds. To systematically investigate the coupling behavior between single LIPSS lines, we produced multiple samples with the same LIPSS periodicity but variations in the spacing between the LIPSS lines (Fig. 4c). It should be noted that for line spacings below 700 nm, the LIPSS formation is not stable anymore (see Supplementary Fig. 4) and therefore not suitable as DFB grating. The area that is pumped is constant for all experiments, with a diameter of roughly 30 µm.

Under the experimental conditions described above, the threshold dependence as a function of spacing between adjacent LIPSS lines provides clear insights into the coupling behavior of the whole system. If the coupling between LIPSS lines is weak or negligible, the excited area would show the same behavior as a single LIPSS line, and we would expect to see a constant threshold, independent of the spacing between LIPSS lines. On the other hand, if coupling is significant, all LIPSS lines within the pumping region would act as one large resonator. Increasing the line spacing between adjacent LIPSS lines reduces the total positive optical feedback and coupling strength[49], resulting in higher thresholds.

Our measurement results, presented in Fig. 4d, show a strongly non-linear threshold behavior as a function of the distance between adjacent LIPSS lines, with a characteristic slope coefficient of about 2 in a double-logarithmic plot, until the threshold saturation regime is reached. This saturation begins at a line distance of 9.3 µm, determined

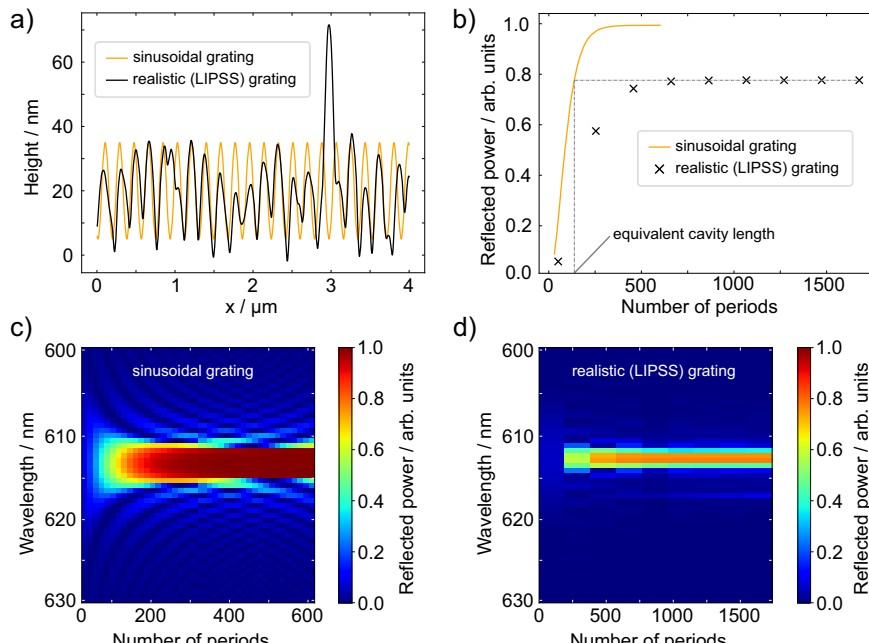

**Fig. 5 | Simulations to estimate the effective mode volume of lasing by comparing reflectivities of perfect and realistic gratings. a** Experimentally measured LIPSS profile and equivalent ideal sinusoidal grating profile. **b** Simulated spectral peak reflectivities for varying grating lengths. The reflectivity of the realistic profile saturates at a significantly lower value than the sinusoidal grating. Comparing the saturation value with the reflectivity of the sinusoidal grating leads to the conclusion that the LIPSS grating can be effectively described as a sinusoidal grating with a length of 28 μm. Contour plots of reflectivity spectra as a function of grating length for **c** ideal and **d** realistic LIPSS structures.

by the intersection of tangents fitted to both regimes, as shown in Fig. 4d. Above this distance, individual LIPSS structures stop interacting and operate as independent laser oscillators. Notably, this threshold distance is of the same order of magnitude as half the pump spot diameter. For a line distance above 15 μm, at most one LIPSS line is effectively excited, given the Gaussian lateral intensity distribution across the diameter of the pump beam. Consequently, the lasing threshold becomes independent of the LIPSS spacing, as neighboring LIPSS structures lie outside the pumped region and cannot contribute to lasing. The results of the last experiment series strongly suggest that coherent cross-coupling among LIPSS lines, responsible for the positive optical feedback in the system, is effectively taking place in our organic thin-film DFB devices.

We have successfully demonstrated the use of LIPSS as optical gratings for organic DFB lasers for the first time. Although resonances in the linear regime are neither observed experimentally nor in simulations, the periodic surface height variation provides sufficient optical feedback along the LIPSS grating to reach the lasing threshold in optically pumped organic DFB structures. This is demonstrated by varying the LIPSS periodicity, which leads to a corresponding shift of the lasing wavelength. Adjacent gratings share the same periodicity and phase, which allows for efficient optical coupling between individual LIPSS lines. We study this coupling by measuring the lasing characteristics, and in particular the lasing thresholds, as the distance between adjacent LIPSS lines varies.

LIPSS grating imperfections limit the mode volumes involved in the lasing process. Using FDTD simulations based on measured 1D LIPSS profiles, we estimate the effective grating length contributing to coherence build-up to be about 28 μm. Experimental measurements of the lasing threshold suggest a value three times smaller, about 10 μm. This discrepancy arises because our simulations ignore the dimension along the LIPSS grating lines (perpendicular to the grating vector), and thus do not account for imperfections in this dimension.

Our LIPSS-based DFB resonators are created in a one-step process, offering unique advantages: high production speed, versatility, and cost-effectiveness. These characteristics make LIPSS ideal for rapid prototyping of DFB lasers, especially in time-sensitive or resource-limited scenarios. The ability to rapidly fabricate complex, large-area photonic systems whilst varying periodicity, grating length, and distance enables comprehensive studies of their optical and lasing properties. This greatly advances our understanding of the field. Moreover, their adaptability to diverse substrates makes LIPSS well-suited for resonators in electrically pumped thin-film lasers, as they can be produced on a wide variety of materials. Another key application is using LIPSS lasers as coherent on-chip light sources for photonic experiments. Coated with a laser dye like $Alq_3$:DCM, these gratings can serve as optically pumped sources of coherent light, eliminating the need for precise external light source coupling and simplifying integration into thin film photonic circuits.

## Methods
### Femtosecond laser micromachining
Femtosecond laser processing was conducted with a micromachining workstation (FemtoLAB, Workshop of Photonics) using a Pharos femtosecond laser from Light Conversion, operating at a 50 kHz repetition rate and emitting 200 fs pulses at a wavelength of 515 nm (second harmonic of the fundamental frequency). The fused silica substrate was mounted on a high-precision 3D translation stage, enabling precise control of its position relative to the laser beam. This setup ensures accurate synchronization of laser pulses in both time and space. To structure the substrate surface, the laser was focused using a microscope objective lens with a numerical aperture (NA) of 0.42 (Mitutoyo M Plan APO NIR 50X).

### DFB sample fabrication
Before laser micromachining, the fused silica substrate was pre-cleaned using acetone and isopropanol in an ultrasonic bath for 5 min each, followed by a 10-minute oxygen plasma treatment. The raster-type scanning was performed by moving the sample from left to right while maintaining a constant pitch distance between adjacent scan lines. The

polarization of the laser beam was set parallel to the laser-driven direction to form laser-induced periodic gratings in the first-order DFB regime. To achieve high-quality DFB-like LIPSS ripples with desired periodicities and minimal phase variations among various raster-scanned groups, specific parameters such as light intensity, repetition rate, and the number of impinging pulses were carefully selected based on the corrugation requirements.

A small molecule organic thin-film system, $Alq_3$:DCM (2 wt%), which exhibits optical gain under optical pumping, was thermally co-evaporated to a thickness of 500 nm onto the corrugated fused silica substrates. Prior to evaporation, the corrugated substrates were re-cleaned with deionized water, acetone, and isopropanol in an ultrasonic bath, followed by a 10-minute oxygen plasma treatment to remove any residual debris.

### Structure characterization

The corrugated fused silica substrates were characterized using scanning electron microscopy (SEM, Gemini 500, Zeiss) to analyze the surface morphology and corrugation profiles. All SEM images were acquired with an electron energy of 3 keV. Prior to SEM imaging, a 5 nm thick Au:Pd (80:20) alloy was sputtered onto the surface to improve conductivity and minimize charging effects. To obtain cross-sectional images of the DFB sample, it was cut using a 30 keV $Ga^+$ focused ion beam (FIB), with a sputtered Au:Pd (80:20) layer of 50 nm and an electron beam deposited platinum coating layer of 400 nm applied on top. Additionally, atomic force microscopy (AFM, CombiScope) was employed with a scanning rate of 0.8 rows/s and an oscillation amplitude of 80 nm, covering a $4 \times 4 \, \mu m$ square area.

### Lasing characterization

The lasing characterization was conducted using a femtosecond regenerative amplifier (Astrella, Coherent), which produces fundamental laser light with a wavelength of 800 nm and a pulse duration of 45 fs. A barium-borate (BBO) crystal was used for second harmonic generation to produce a 400 nm laser beam, matching the matrix absorption peak of $Alq_3$ at around 390 nm, to excite the DFB sample. The sample was mounted on a translation stage, precisely positioned by two step motors, allowing for highly accurate placement at any arbitrary location for measurements. Emission spectra from the DFB sample were collected and recorded perpendicular to the sample surface using a charge-coupled device (CCD) spectrometer with a resolution of 0.2 nm.

### Reporting summary

Further information on research design is available in the Nature Portfolio Reporting Summary linked to this article.

## Data availability

The data supporting the findings of this study are available in this article (including the source data of each Figure) and its Supplementary Information files. All data are available from the corresponding author upon request. Source data are provided with this paper.

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

## Acknowledgements

This work was supported by the Deutsche Forschungsgemeinschaft (DFG)-project No. LE 747/68-1 (project-ID 442597684, "Microcavity Lasers") (M.S.). The authors acknowledge the use of the facilities in the Dresden Center for Nanoanalysis at the Technische Universität Dresden. Furthermore, the authors thank for the financial support from the DFG through the Würzburg-Dresden Cluster of Excellence on Complexity and Topology in Quantum Matter - ct.qmat (EXC 2147, project ID: 39085490) (J.L.). J.B. gratefully acknowledges funding from the Federal Ministry of Research, Technology and Space (BMFTR) as part of the project to establish the German Center for Astrophysics (03WSP1745).

## Author contributions

The first two authors (T.D. and T.A.) were the lead writers of the paper and contributed equally to the work. T.D. fabricated the first LIPSS samples and did the experiments on LIPSS periodicity and raster line distance together with M.S., T.A. conducted the experiments on the stripe length variation. F.P., with help from J.L., carried out the reflectivity simulations. M.L. performed sample cutting with FIB and the cross-section measurement with SEM. J.B. and B.R. coordinated the LIPSS lasing project. K.L. motivated this work and directed the project.

## Funding

## Competing interests

The authors declare no competing interests.
