## [Transparent Peer Review file · Nature Communications]

Read my LIPSS: organic lasers on micromachined resonators

Corresponding Author: Mr Tobias Antrack

Version 0:

Reviewer comments:

Reviewer #1

(Remarks to the Author)

The manuscript entitled "Read my LIPSS: organic lasers on micromachined resonators" deals with the generation and utilization of laser-induced periodic surface structures (LIPSS) as laser optical resonators in the form of first-order distributed feedback (DFB) Bragg gratings. The formation process of LIPSS in different glasses has been investigated in various studies, with varying degrees of success, as the non-linear interaction of the laser radiation with the dielectric material presents an enormous hurdle. For this reason, the manuscript is generally very exciting and offers a novel extension of the application of LIPSS. However, some aspects still require intensive discussion and clarification, which is why I cannot propose the manuscript for publication in its current form.

The main point of the discussion lies in apparently different laser-induced structures, which are summarized under the term high-spatial frequency LIPSS (short HSFL), but in my opinion have different origins.

I agree that the structures shown in Figure S1 represent HSFL, especially since their orientation varies with the direction of the linear beam polarization. By the way, an alignment perpendicular to the polarization, as claimed in the introduction, is not always given, but depends on the specific type of material (see Bonse et al. (2012) <https://doi.org/10.2351/1.4712658> DOI). A closer look to Figure S2 reveals structures that are characterized by slight curvature corresponding to the circular focal spot of the processing beam. It appears that these are structures created by diffraction at the rim(s) formed by the prepulse. This effect was studied by Ben-Yakar et al. on borosilicate glass and published in 2003 (<https://doi.org/10.1063/1.1619560>). One indication of this is the coincidence/increase in the spatial period with the feed rate (i.e. the pulse-to-pulse spacing) - a phenomenon that is not actually observed in HSFL.

Regarding the review process, it is not possible to clarify whether these are actually the same type of structures due to the very low resolution of the images provided. There is also a lack of information, for example on the fluence in Figure S1 or generally on the focal spot size.

However, a possible experiment to refute the rim theory during the formation of the structures in Figure S2 would be a line scan with linear polarization rotated by 90° (here the structures would have to rotate as well). Alternatively, the direction of the scan line could be changed (from left to right and vice versa), which would have to lead to the opposite curvature of the "HSFL".

Due to the originality of the work, I suggest resubmission of the paper after clarification and a major revision.

Reviewer #2

(Remarks to the Author)

Reviewer #3

(Remarks to the Author)

The authors present a well-done study of femtosecond laser fabrication of LIPSS structures on glass followed by

measurements of the efficiency of coupling between LIPSS distributed feedback structures through the use of pulsed laser optically pumped lasing observed in Alq3:DCM composites.

This is solid scientific work with sound methodology, a sufficiently detailed description and reasonable conclusions. The approach does appear to be an excellent way of investigating coupling between LIPSS structures.

This said, the primary purpose of the research is somewhat unclear. If it was to show a path towards a successful solid state organic laser, that path is primarily dictated by the physics of the excitation and luminescence in the candidate system, not the resonator geometry. If it was to show that resonantly coupled LIPSS structures can be efficiently and reproducibly fabricated with femtosecond lasers then a discussion of the primary applications of such resonators should be included.

Reviewer #4

(Remarks to the Author)

Review on : Read My LIPSS : Organic lasers on micromachined resonators

by Tiange Dong* Tobias Antrack*, Frithjof Pietsch, Jakob Lindenthal, Markus Löffler Bernd Rellinghaus, Johannes Benduhn, Markas Sudzius, and Karl Leo

The paper entitled « Read My LIPSS : Organic lasers on micro-machined resonators » present experimental results on optically pumped organic lasers that uses distributed feedback cavities fabricated using laser-induced micro-machining. The technique used to pattern the cavity is known as Laser-Induced Periodic Surface Structures (LIPSS). The paper is composed of 5 sections : 1- The fabrication of LIPSS-based cavity, 2- The device fabrication, 3- the demonstration of lasing action under optical pumping, 4- The analysis of the laser threshold as a function of geometry, 5-Coherent coupling between different cavities

The novelties are:

1. As mentioned by authors (Ref 24-28) the fabrication of DFB cavity using the LIPSS technique is not really new but it has mainly been used to pattern fibers (Fiber Bragg gratings - FBG) for use in the infrared region (thus for larger periods). Only, reference 27 deals with glass sample. Thus, a first originality of the current work is in the use of LIPSS to fabricate photonic structures in the visible range which is more difficult to achieve than in the infrared because of smaller periodicity. A second originality is in the application of the LIPSS technique to fabricate organic lasers which is quite a different field from the FBGs.

2. The lateral coherent coupling between organic DFB lasers

Significance of the paper

1. The high throughput of the LIPSS technique allow easy experiments on cavity geometries that provide useful and significant information to the organic laser field. Indeed, although Alq3:DCM as laser gain organic materials is very common, this paper provides very interesting and useful information on how the laser threshold is related to the cavity length, the quantification of the critical cavity length for building lasing, the critical length to achieve maximum reflectivity, and a relatively easy link between experiments and the coupled wave theory. This will be very helpful for the design of optimal DFB cavities which is key for the organic laser diode quest.

Another important significance of this work is what it bring to the solid state organic laser as the established literature does not contain enough material on the optimal design of laser cavity for organic laser and ore specifically tha could be useful for the organic diode laser topic.

Therefore, I would recommend it for publication after the following comments have been made. Indeed, the article can be improved with changes that will facilitate reading and understanding of the readers

A. Check for possible typo errors :

1. p5 line 129: ...the direction of the resulting.... ?

2. p5 line 132: "No grating_ can be created" ?

B. Some figures could be improved

3. Fig 1.c: This SEM image is one of the main results. It is supposed to help the reader (eventually new to the field) to really figure out what LIPSS looks like. But in its current form, it is too small and the details can hardly be distinguished. Such a figure deserve to be much larger in the main paper and with a better contrast. In the supplementary material, the figures are slightly better, still there is room for improving them with better contrast or larger images since there is no matter with number of pages in it.

4. Same remarks for SEM images Fig 4 a, and c that are too small

5. In Fig 1.d and e, the x axis is displayed in wave number (nm⁻¹), while in the supplementary materials similar figures (S4 b, c, e and f) are displayed in ("spatial"?) frequency but the unit is (nm⁻¹). More consistency in the edition would make the reading easier. Consider adding the word "spatial" in front of frequency.

6. Check possible inconsistencies also in Fig 1.e and Fig 4.c and f.

7. Figure 3.a and the inset show emitted spectra with increasing pumping intensity. However it is very hard for the reader to really distinguish or even get an approximation of the FWHM, nor is it possible to figure out for which pump intensity it has been obtained. A larger figure and inset would improve the paper.

8. Fig 5. a shows the amplitude of the corrugations which is a key parameter of the DFB cavities, but it is partly hidden by the orange curve. Improving the visibility of the dark curve (Realistic LIPSS grating profile) would help the readers to access the non-uniformity and the amplitude of the corrugation.

9. Fig 5.b, c and d: x axis is displayed in grating length / period. This might be confusing ? There is possible confusion: (Length of the cavity in μm divided by the period in μm → number of periods ?) or (Gating length expressed in terms of period ?). This casts doubt on other figures which doesn't give justice to the experimental results. In fig 5.a the x-axis is $X/\mu\text{m}$

whereas it is in apparently in μm ($4\mu\text{m}/21$ period gives 190nm which is indeed approximately the period of the grating). Consider using a different symbol than “ μ ” to express units...; $[\mu\text{m}]$ or for example (height in nm) (Grating length in period) or simply (number of periods). For other figures, the risk of confusion is less, but consistencies among the figures is important.

C. Some key data or information are missing in the main paper although some of them are presents in the supplementary materials)

10. The amplitude of the ripples is not explicitly indicated apart from the supplementary material (Table S1), whereas it is key in cavity design. This data is really missing to the reader in the first part of the document where the pattern is described. It can also be added on figure 2.c where a vertical scale would be usefull. Also explicit indications of both the amplitude of the ripples to be compared to the thickness of the organic layer (indicated line 172).

11. The width of the stripes is not indicated. It is usefull for the calculation of the mode volume.

12. Line 218-222: the discussion is about the wave-guided mode. In waveguided mode the effective index of refraction is indeed lower than the refractive index so the reader is not surprise by the findings

1. A plot of the electric field distribution of the different wave-guided modes inside a 500nm thick waveguide (without grating) would help the reader to better visualize the configuration and understand the discrepancy between the different refractive indices. Adding in the Supplementary Materials, a figure relevant for the calculation of the volume of mode mentioned later in the paper.. For example, <https://www.sio.eu/oms.html> or other simulation softaare or applet.

13. What are the hot quality factors of the different cavities ? Is it possible to estimate the cold quality factors from the simulation?

1. In the current form of figure 3, the reader cannot estimate Q factors from the FWHM in Fig 3.b. because the right y-axis lacks of accuracy close to zero. A log scale would help. An indication of the Q factor somewhere in the text would help the reader. How the Q factor changes when the length of he cavity is changed.

D. Questions related to the scientific content

14. How does the corrugation amplitude as small as $\sim 30\text{nm}$ while the organic layer thickness is 500nm, affect the reflectivity and the lasing action?

1. Does the upper corrugation at the organic/air interface play a role in the laser threshold ?

2. Does the phase difference of roughly π between the bottom and top layer modification play a role in the threshold and in the quality factor ?

15. Is it possible to enhance the ripple amplitude ? Would it help to improve the reflectivity or the quality factor ?

16. Line 120 indicates “we did not notice a dependency of Λ on the laser fluence (see Figure S2)”. Does the amplitude of the corrugation depends on the laser fluence ?

17. Does the Fourier analysis indicated line 29-30 of Fig S2 show changes in amplitude when the fluence is changed from 126 nJ, to 135 nJ, and 144 nJ). ? (Amplitudes of the FFT plots) as an indication of the change in amplitude of the ripples.

18. Line 138 indicates a FFT was performed on the top-view of the SEM image. Could you indicates the way it has been performed ? Is the FFT performed on the entire image or on stripes centered on a particular grating, or even a line in the center of a particular grating ?

19. Line 145-14: Does a 1D FFT calculated on a band (not a single line) of the image centered on a grid width equal to the grid width provide useful information about the non-uniformity of ripples, including the difference between the center and edges of the network?

20. Line 255-256: “Surface profile disturbance affect the optical quality due to spectral noise onset”: How the lateral variation of the stripe induce noise ? Would a 2D FF analysis help to answer this issue ?

21. Line 218-222: Regarding the wave-guided modes:

1. How many modes co-exist in a 500nm thick organic layer ?

2. What are the values of the effective refractive index of the different waveguided modes?

3. Do they fit with the calculated ≈ 1.64 ?

22. Line 275: The concept of Coherence propagation is unclear.

23. Page 10 Line 295:; The legend of figure 5 appears unclear: “Simulations to estimate the effective mode volume of lasing.” How do the different figure relate to the effective mode volume ? How do you use the dimension of the grating, and the field distribution of the different modes (Q 21) to calculate the mode volume.

24. Page 10 also line 297: Legend of Figure 5.b use the expression “peak reflectivity” which is unclear. Why is it peak ? Does it relates to the reflectivity saturation levels line 290 ? Does the reflectivity saturation level have the same meaning than the peak reflectivity ?

25. Line 295: The legend of figure 5 starts with “Simulations to estimate the effective mode volume of lasing.”. How do the figures 5 provide evidence of the calculation of the mode volume. I gives indications on some of the dimensions (cavity width is absent), so how does it allow to calculate the mode volume ?

Version 1:

Reviewer comments:

Reviewer #1

(Remarks to the Author)

The authors responded to all my suggestions and questions in a very detailed and thorough response letter. This also applies to the other reviewers. Due to the interesting topic, I propose the revised manuscript for publication.

Reviewer #2

(Remarks to the Author)

Reviewer #4

(Remarks to the Author)

Dear Authors,

Your detailed answers to my comments and questions about your manuscript entitled "Read my LIPSS: organic lasers on micromachined resonators" are convincing. The changes made to the manuscript greatly clarified the points I had raised and improved the quality of the article.

In particular it is now much easier for the reader to evaluate the strengths and limits of the LIPSS method for the fabrication of DFB cavity for organic lasers.

I am now convinced that your manuscript meets the criteria for publication in Nature Com and I recommend it for publication. I think your work will contribute to the field of organic lasers.

Sincerely,

Rebuttal letter accompanying our re-submission of:

"Read my LIPSS: Organic lasers on micromachined resonators" (Nature Communications, NCOMMS-25-05710)

Dear Editor, dear Reviewers,

Thank you for the thorough peer-review of our manuscript and the very helpful criticism. In the document below, we comment on all the concerns and points raised by the reviewers. For easy readability, we visually structure our response as follows:

The comments and points of the referees are copied in black color.

Our response is colored in blue and is indented to the right.

Furthermore, we cite parts from the revised manuscript or the revised supporting information to afford a quick impression of the improvements and changes made to the text. These citations are indicated by green text and are further indented to the right.

We hope that we fully answered all questions and hope that the manuscript is now ready for publication.

Sincerely,
Tiange Dong and Tobias Antrack (on behalf of all co-authors)

Table of Contents

Reviewer #1:.....	2
Reviewer #2:.....	15
Reviewer #3:.....	15
Reviewer #4:.....	16
References for the rebuttal letter:	52

Reviewer #1:

The manuscript entitled "Read my LIPSS: organic lasers on micromachined resonators" deals with the generation and utilization of laser-induced periodic surface structures (LIPSS) as laser optical resonators in the form of first-order distributed feedback (DFB) Bragg gratings. The formation process of LIPSS in different glasses has been investigated in various studies, with varying degrees of success, as the non-linear interaction of the laser radiation with the dielectric material presents an enormous hurdle. For this reason, the manuscript is generally very exciting and offers a novel extension of the application of LIPSS. However, some aspects still require intensive discussion and clarification, which is why I cannot propose the manuscript for publication in its current form.

We thank the reviewer for the positive evaluation and pointing out that the topic and our results are very exciting. We hope, our answers will satisfy the Reviewer's concerns.

1. The main point of the discussion lies in apparently different laser-induced structures, which are summarized under the term high-spatial frequency LIPSS (short HSFL), but in my opinion have different origins.

I agree that the structures shown in Supplementary Figure 1 represent HSFL, especially since their orientation varies with the direction of the linear beam polarization. By the way, an alignment perpendicular to the polarization, as claimed in the introduction, is not always given, but depends on the specific type of material (see Bonse *et al.* (2012) <https://doi.org/10.2351/1.4712658> DOI).

We thank the reviewer for this input. The mentioned paper by Bonse *et al.*¹ was already cited in the original manuscript. However, we decided to additionally point out that perpendicular polarization is not always given for HSFL:

HSFL is characterized by spatial periods significantly smaller than the irradiation wavelength and orientations perpendicular to the polarization for most materials¹.

To further clarify that, at least for glass, HSFL is known to be perpendicularly polarized to the laser polarization, we improved the following sentence in the manuscript:

With the structures having periodicities significantly below the structuring laser wavelength, and polarization being parallel to the scanning direction (and consequently perpendicular to the resulting LIPSS ripples, see Supplementary Figure 1), we are in the regime of high-spatial frequency LIPSS (HSFL) for glass as the substrate^{2,3}.

2. A closer look to Supplementary Figure 2 reveals structures that are characterized by slight curvature corresponding to the circular focal spot of the processing beam. It appears that these are structures created by diffraction at the rim(s) formed by the prepulse. This effect was studied by Ben-Yakar *et al.* on borosilicate glass and published in 2003 (<https://doi.org/10.1063/1.1619560>). One indication of this is the coincidence/increase in the spatial period with the feed rate (i.e. the pulse-to-pulse spacing) - a phenomenon that is not actually observed in HSFL.

We thank the Reviewer for pointing out this discrepancy from our claim of having LIPSS structures.

First, we want to emphasize that the curved feature of the ripples only appears for rather high pulse fluence, whereas for lower pulse fluences (70 nJ in Supplementary Figure 2a and 86 nJ in Supplementary Figure 2b), there is no curvature observed but straight lines instead.

Second, the correspondence of the curvature is shown in Rebuttal Figure 1, with the diameter of the beam spot clearly indicating that, for relatively high pulse fluence, we indeed do not have LIPSS but observe an effect similar to Ben-Yakar *et al.*⁴. We like to point out that the dimensions in our case are more than one order of magnitude smaller than in the mentioned publication. Additionally, the distance between the ripples and the width of the line are significantly smaller than the laser spot size (189 nm periodicity with 1.1 μm beam waist). For lower pulse fluence, there is no curvature of the rims, hence it appears very similar to LIPSS.

Rebuttal Figure 1 | Surface ripples fabricated with high fluence of 104 nJ per pulse. The red circles represent the beam spot size of 1.1 μm . It can be seen that the curvature of the ripples corresponds to the focus spot size of the structuring laser.

A literature research on the problem revealed that control of LIPSS periodicity by changing the pulse-to-pulse spacing is reported in multiple publications⁵⁻⁷. All of them observed increased LIPSS periodicity with increased pulse-to-pulse spacing. Here, the work from Sun *et al.*⁶ is of special interest because they classify HSFL in 3 regimes (see Rebuttal Figure 2). The reported Regime III fits well to our observed relation of pulse spacing to grating period, which is described as point-by-point writing instead of nanograting formation. For higher pulse fluences, we observe very similar behavior, but for lower pulse fluences, we observe typical HSFL behavior (see response to question 4). In contrast to Sun *et al.*⁶, we observe a transition between Regime II and III depending on laser fluence.

[REDACTED]

Rebuttal Figure 2 | This figure is taken from Sun *et al.*⁶. Original caption: [REDACTED]

In summary, the Reviewer is correct about his observation that not all presented nanostructures can actually be classified as LIPSS. However, we also report on LIPSS with the same periodicity as the shot-to-shot spacing, and most importantly for this work, they also serve well as a 1st order DFB Bragg grating. We will add the following discussion to point that out:

Both observations indicate that rim-formation⁴² due to single point writing³⁹ is also involved in the grating formation (see Supplementary Note 1 for extended analysis).

3. Regarding the review process, it is not possible to clarify whether these are actually the same type of structures due to the very low resolution of the images provided. There is also a lack of information, for example on the fluence in Supplementary Figure 1 or generally on the focal spot size.

We thank the Reviewer for pointing out the missing information. We apologize for the low image resolution. We have now increased the resolution of the pictures and hope, that this will result in better details.

The focus spot in our setup (a 515 nm laser and an infinite-corrected objective with 50x magnification) has a diameter of 1.1 μm . This was measured via the knife-edge method with the beam diameter being the lateral distance between the point with 10% and the point with 90% intensity⁸. We mention the focus spot size now in the main text:

With a laser focus point size of 1.1 μm and a pulse fluence of around 80 nJ, the energy density during fabrication is comparable to previously reported threshold energy to fabricate LIPSS⁴².

Additionally, we mention the applied laser pulse fluence of the structures shown in Supplementary Figure 1:

Supplementary Figure 1 | Characterization of the laser polarization dependency in the dynamic structuring approach. The applied laser polarization angles are a) $\theta = 0^\circ$, b) $\theta = 22.5^\circ$, c) $\theta = 30^\circ$, d) $\theta = 60^\circ$, and e) $\theta = 90^\circ$, represented by the inset arrow lines. The pulse fluences for a-c) were 82 nJ, and 78 nJ for d) and e). The SEM images of the resulting structures clearly show that the orientation of linear-scanned laser-induced nano-gratings is perpendicular to the incoming laser polarization.

4. However, a possible experiment to refute the rim theory during the formation of the structures in Supplementary Figure 2 would be a line scan with linear polarization rotated by 90° (here the structures would have to rotate as well). Alternatively, the direction of the scan line could be changed (from left to right and vice versa), which would have to lead to the opposite curvature of the "HSFL".

We thank the reviewer for the proposed experiment. We fabricated multiple samples consisting of horizontal and vertical lines where the same parameters were applied for corresponding pairs. In the following, we will discuss our observations and if our structures can be classified as LIPSS. The presented SEM scans show pairs of samples, where for each pair the travel direction was changed but all other parameters were kept constant. With this approach we want to examine if the ripple orientation is staying constant (independent of travel direction) and hence only depends on the laser polarization direction, or the ripple orientation is changing according to the travel direction, which will indicate the influence of the single pulse writing mechanism⁶. The laser polarization direction was always horizontal.

For pulse-to-pulse spacings below the range used for our Bragg laser resonators, we can easily demonstrate HSFL behavior (the LIPSS orientation does not change when the travel direction is changed), as shown in Rebuttal Figure 3:

Rebuttal Figure 3 | LIPSS formation with small spot-to-spot spacings | The first two samples (a-d) were fabricated with a pulse-to-pulse spacing of 111 nm whereas the last sample (e-f) was fabricated with a pulse-to-pulse spacing of 143 nm. The structuring laser polarization was kept horizontal for all samples, only the travel direction was changed. The formation of HSFL can be clearly demonstrated by the orientation of the ripples being always vertical and hence perpendicular to the laser light polarization.

For a pulse-to-pulse spacing of 189 nm as used for creating lasing resonators, the clarification of the structures is not so straightforward (see Rebuttal Figure 4). It can be seen that the major ripple orientation is rotating depending on the travel direction which indicates the absence of clear LIPSS behavior but rather a point-by-point writing mechanism as described by Sun *et al.*⁶ in Rebuttal Figure 2. When looking at the

detailed shape of the rims, small sub-ripples on the rims themselves can be observed, which consequently have an orientation perpendicular to the laser polarization direction (as expected from HSFL on glass). Additionally, there are irregularities in the structures where the main ripple orientation locally changes and is aligned perpendicular to the structuring laser polarization (as expected from LIPSS). This implies that we are in the transition zone between LIPSS formation and point-by-point writing. Due to the parallel alignment of the ripples formed by point-by-point writing and the ripples formed by LIPSS, this transition zone is not easily observable when the scanning direction is parallel to the laser polarization like during the fabrication process of our laser resonators.

Rebuttal Figure 4 | Fabrication conditions similar to DFB resonator fabrication. These structures were fabricated with a pulse-to-pulse spacing of 189 nm and a pulse fluence of 80 and 85 nJ, which corresponds to the most used condition to fabricate optical gratings. The vertical lines show clear horizontal ripples, which points towards single pulse writing as the major effect involved in the ripple formation. However, closely looking at the darker defect spots, it can be seen that the ripple orientation changes locally. In these spots, LIPSS formation seems to have a higher influence on the ripple formation direction than the single pulse writing process.

Going to very low laser fluences during fabrication allows us to reach a state where the LIPSS formation is more pronounced than the point-by-point writing in the case of the travel direction being perpendicular to the laser polarization (see Rebuttal Figure 5).

Rebuttal Figure 5 | LIPSS-formation at low laser pulse fluence. All structures were fabricated with a pulse-to-pulse spacing of 189 nm, like it was used for the laser resonator structures. For the lower fluence, clear vertical lines for the vertical travel direction can be seen, which clearly hints towards LIPSS formation. With increasing laser pulse fluence, additional horizontal ripples are distinguishable, which are likely to be

combing from rim formation. This indicates that with our grating periodicity of 189 nm, we are in the transition zone between LIPSS and single pulse writing, and both processes are involved in the grating formation.

To summarize, the Reviewer is right about his doubts that some of the presented nano-gratings were solely formed by LIPSS. However, we can show that LIPSS is still appearing within the range of our fabrication parameters. Since for our grating fabrication, the ripple formation direction from both mechanisms (LIPSS and single point writing) coincides, we assume that both effects play a role in the ripple formation. According to the work by Sun *et al.*⁶, we are in the transition zone between Reg. II and Reg. III, which corresponds to our observations.

The exact mechanisms involved in the fabrication of the surface nanogratings are not of relevance for the application as lateral Bragg resonators for thin-film lasers. The structures fabricated with rather high pulse fluence tend to have a more sinusoidal height profile which is beneficial for the reflectivity of the grating. But when the pulse fluence is further increased and the ripples appear with notable curvature, the reflectivity is lowered and hence the observed lasing threshold higher. In that case, the phase differences between the center and the edge of the grating partly cancel out the reflected wave.

We added this sentence to the description of the LIPSS formation in the main text:

Both observations indicate that rim-formation⁴² due to single point writing³⁹ is also involved in the grating formation (see Supplementary Note 1 for extended analysis).

And to the Supplementary Information, we added the following discussion:

Supplementary Note 1 – Extended analysis of the processes involved in the ripple formation

Ripple formation due to LIPSS should result in a ripple orientation perpendicular to the laser polarization direction and independent of the travel direction. However, the process of single point writing will result in ripples parallel to the travel direction and independent of the laser polarization direction. Therefore, we fabricated multiple samples consisting of horizontal and vertical lines where the same parameters were applied for corresponding pairs. In the following, we will discuss our observations and if our structures can be classified as LIPSS. The presented SEM scans show pairs of samples, where for each pair the travel direction was changed but all other parameters were kept constant. With this approach we want to examine if the ripple orientation is staying constant (independent of travel direction) and hence only depends on the laser polarization direction, or the ripple orientation is changing according to the travel direction, which will indicate the influence of the single pulse writing mechanism³. The laser polarization direction was always horizontal.

For pulse-to-pulse spacings below the range used for our Bragg laser resonators, we can easily demonstrate HSFL behavior (the LIPSS orientation

does not change when the travel direction is changed), as shown in Supplementary Figure 13:

Supplementary Figure 13: LIPSS formation with small spot-to-spot spacings | The first two samples (a-d) were fabricated with a pulse-to-pulse spacing of 111 nm whereas the last sample (e-f) was fabricated with a pulse-to-pulse spacing of 143 nm. The structuring laser polarization was kept horizontal for all samples, only the travel direction was changed. The formation of HSFL can be clearly demonstrated by the orientation of the ripples being always vertical and hence perpendicular to the laser light polarization.

For a pulse-to-pulse spacing of 189 nm as used for creating lasing resonators, the clarification of the structures is not so straightforward (see Supplementary Figure 14). It can be seen that the major ripple orientation is rotating depending on the travel direction which indicates the absence of clear LIPSS behavior but rather a point-by-point writing mechanism as described by Sun *et al.*³. When

looking at the detailed shape of the rims, small sub-ripples on the rims themselves can be observed, which consequently have an orientation perpendicular to the laser polarization direction (as expected from HSFL on glass). Additionally, there are irregularities in the structures where the main ripple orientation locally changes and is aligned perpendicular to the structuring laser polarization (as expected from LIPSS). This implies that we are in the transition zone between LIPSS formation and point-by-point writing. Due to the parallel alignment of the ripples formed by point-by-point writing and the ripples formed by LIPSS, this transition zone is not easily observable when the scanning direction is parallel to the laser polarization like during the fabrication process of our laser resonators.

Supplementary Figure 14: Fabrication conditions similar to DFB resonator fabrication |

These structures were fabricated with a pulse-to-pulse spacing of 189 nm and a pulse fluence of 80 and 85 nJ, which corresponds to the most used condition to fabricate optical gratings. The vertical lines show clear horizontal ripples, which points towards single pulse writing as the major effect involved in the ripple formation. However, closely looking at the darker defect spots, it can be seen that the ripple orientation changes locally. In these spots, LIPSS formation seems to have a higher influence on the ripple formation direction than the single pulse writing process.

Going to very low laser fluences during fabrication allows us to reach a state where the LIPSS formation is more pronounced than the point-by-point writing in the case of the travel direction being perpendicular to the laser polarization (see Supplementary Figure 15).

Supplementary Figure 15: LIPSS-formation at low laser pulse fluence | All structures were fabricated with a pulse-to-pulse spacing of 189 nm, like it was used for the laser resonator structures. For the lower fluence, clear vertical lines for the vertical travel direction can be seen, which clearly hints towards LIPSS formation. With increasing laser pulse fluence, additional horizontal ripples are distinguishable, which are likely to be combing from rim formation. This

indicates that with our grating periodicity of 189 nm, we are in the transition zone between LIPSS and single pulse writing, and both processes are involved in the grating formation.

To summarize, LIPSS is not always the main mechanism responsible for the ripple formation. However, we can show that LIPSS is still appearing within the range of our fabrication parameters. Since for our grating fabrication, the ripple formation direction from both mechanisms (LIPSS and single point writing) coincides, we assume that both effects play a role in the ripple formation. According to the work by Sun *et al.*³, we are in the transition zone between Reg. II and Reg. III, which corresponds to our observations.

The exact mechanisms involved in the fabrication of the surface nanogratings are not of relevance for the application as lateral Bragg resonators for thin-film lasers. The structures fabricated with rather high pulse fluence tend to have a more sinusoidal height profile which is beneficial for the reflectivity of the grating. But when the pulse fluence is further increased and the ripples appear with notable curvature, the reflectivity is lowered and hence the observed lasing threshold higher. In that case, the phase differences between the center and the edge of the grating partly cancel out the reflected wave.

To minimize the risk of future readers to be confused by the curvature notable in Figure 4a in the main text, we replaced the image with a grating which does not show such significant curvature:

Fig. 4 | Coupling behavior and effective mode volume. **a** SEM images of LIPSS stripes with constant line separation but varying stripe length, and **b** the measured lasing thresholds for different stripe lengths. The threshold is decreasing non-linearly with increasing stripe length until saturation is reached. **c** SEM image of a LIPSS-structured glass substrate and exemplary representation of the area being excited by the pumping beam. **d** Lasing threshold as a function of the raster line distance d_L between adjacent LIPSS lines for a fixed pump spot size of $30 \mu\text{m}$. The threshold reaches a saturation just before d_L is around half of the pump spot diameter D . For $d_L > D/2$ effectively only one LIPSS line is being excited. The dashed lines are linear fits of the measurement data.

Due to the originality of the work, I suggest resubmission of the paper after clarification and a major revision.

We thank the reviewer for thoroughly reading our manuscript and giving helpful comments that helped us to improve the quality of the paper. We hope that our answers resolved the Reviewer's questions.

Reviewer #2:

We thank the Reviewer for contributing to the Review!

Reviewer #3:

The authors present a well-done study of femtosecond laser fabrication of LIPSS structures on glass followed by measurements of the efficiency of coupling between LIPSS distributed feedback structures through the use of pulsed laser optically pumped lasing observed in Alq3:DCM composites.

This is solid scientific work with sound methodology, a sufficiently detailed description and reasonable conclusions. The approach does appear to be an excellent way of investigating coupling between LIPSS structures.

We thank the Reviewer for this positive feedback.

This said, the primary purpose of the research is somewhat unclear. If it was to show a path towards a successful solid state organic laser, that path is primarily dictated by the physics of the excitation and luminescence in the candidate system, not the resonator geometry. If it was to show that resonantly coupled LIPSS structures can be efficiently and reproducibly fabricated with femtosecond lasers then a discussion of the primary applications of such resonators should be included.

We thank the Reviewer to point out this uncertainty since the main message of a publication is a major overall aspect. As mentioned by the Reviewer, our approach is not expected to lead the way towards a successful solid state organic laser because other drawbacks would need to be addressed. This is especially valid since the lasing threshold of our optically pumped lasers is higher than with previously reported alternative feedback structures.

However, the extreme ease of fabricating the resonator structures is unprecedented. This opens new use cases like LIPSS lasers as coherent light sources for photonic experiments on a lab scale: coherent light must not be precisely coupled into the structure from an external light source. Instead, the area with the LIPSS resonator and the organic gain material simply needs to be optically pumped from any direction.

To better address the message of the paper, we added this sentence to the introduction:

Using our presented one-step direct laser writing technique allows for the rapid fabrication of a wide variety of model photonic systems, enabling a comprehensive study of their optical and lasing properties.

And updated this sentence in the conclusion to better highlight possible use-cases:

The ability to rapidly fabricate complex, large-area photonic systems whilst varying periodicity, grating length, and distance enables comprehensive studies of their optical and lasing properties.

Reviewer #4:

Review on: Read My LIPSS: Organic lasers on micromachined resonators by Tiange Dong* Tobias Antrack*, Frithjof Pietsch, Jakob Lindenthal, Markus Löffler Bernd Rellinghaus, Johannes Benduhn, Markas Sudzius, and Karl Leo

The paper entitled « Read My LIPSS: Organic lasers on micro-machined resonators » present experimental results on optically pumped organic lasers that uses distributed feedback cavities fabricated using laser-induced micro-machining. The technique used to pattern the cavity is known as Laser-Induced Periodic Surface Structures (LIPSS). The paper is composed of 5 sections: 1 - The fabrication of LIPSS-based cavity, 2 - The device fabrication, 3 - the demonstration of lasing action under optical pumping, 4 - The analysis of the laser threshold as a function of geometry, 5 - Coherent coupling between different cavities

The novelties are:

1. As mentioned by authors (Ref 24-28) the fabrication of DFB cavity using the LIPSS technique is not really new but it has mainly been used to pattern fibers (Fiber Bragg gratings - FBG) for use in the infrared region (thus for larger periods). Only, reference 27 deals with glass sample. Thus, a first originality of the current work is in the use of LIPSS to fabricate photonic structures in the visible range which is more difficult to achieve than in the infrared because of smaller periodicity. A second originality is in the application of the LIPSS technique to fabricate organic lasers which is quite a different field from the FBGs.
2. The lateral coherent coupling between organic DFB lasers

Significance of the paper

1. The high throughput of the LIPSS technique allow easy experiments on cavity geometries that provide useful and significant information to the organic laser field. Indeed, although Alq3:DCM as laser gain organic materials is very common, this paper provides very interesting and useful information on how the laser threshold is related to the cavity length, the quantification of the critical cavity length for building lasing, the critical length to achieve maximum reflectivity, and a relatively easy link between experiments and the coupled wave theory. This will be very helpful for the design of optimal DFB cavities which is key for the organic laser diode quest.

Another important significance of this work is what it brings to the solid-state organic laser as the established literature does not contain enough material on the optimal design of laser cavity for organic laser and more specifically what could be useful for the organic diode laser topic.

Therefore, I would recommend it for publication after the following comments have been made. Indeed, the article can be improved with changes that will facilitate reading and understanding of the readers

We thank the Reviewer for this positive summary of our manuscript. In the following, we addressed all comments and questions raised by the Reviewer.

A. Check for possible typo errors :

1. p5 line 129: ...the direction of the resulting.... ?
2. p5 line 132: "No grating_ can be created" ?

We thank the Reviewer very much for pointing out these mistakes. We corrected and marked them in the manuscript:

Schematic of femtosecond laser scanning to produce laser-induced periodic surface structures (LIPSS), where the direction of the resulting LIPSS grating lines is perpendicular to the polarization of the incident laser beam.

In regime B, the periodicity equals the pulse spacing, whereas in regime C, no **grating** can be created.

B. Some figures could be improved

3. Fig 1.c: This SEM image is one of the main results. It is supposed to help the reader (eventually new to the field) to really figure out what LIPSS looks like. But in its current form, it is too small and the details can hardly be distinguished. Such a figure deserve to be much larger in the main paper and with a better contrast. In the supplementary material, the figures are slightly better, still there is room for improving them with better contrast or larger images since there is no matter with number of pages in it.

We thank the Reviewer for the suggestion of highlighting Figure 1c. We increased the size and show a slightly smaller section of the image to make the LIPSS structure better visible:

Fig. 1 | Fabrication of LIPSS. **a** Schematic of femtosecond laser scanning to produce laser-induced periodic surface structures (LIPSS), where the direction of the resulting LIPSS grating lines is perpendicular to the polarization of the incident laser beam. **b** Dependency of resulting LIPSS periodicity on the lateral structuring laser pulse distance. The resulting LIPSS periodicity is larger than the laser pulse spacing in regime A. In regime B, the periodicity equals the pulse spacing, whereas in regime C, no grating can be created. Regime D denotes gratings produced by ablation (writing grating lines directly with the laser). **c** Exemplary top view SEM image with the green line marking the pixels analyzed for periodicity and phase of the LIPSS gratings. **d** Amplitudes of the Fourier spectrum of 13 analyzed LIPSS gratings with a lateral distance of 1.47 μm and **e** the corresponding Fourier phases. See Supplementary Figures 5 and 8 for more details.

4. Same remarks for SEM images Fig 4 a, and c that are too small

We followed the suggestions of the Reviewer and increased the size of the subfigures 4a and c to make it easier for the reader to understand the sample layouts:

5. In Fig 1.d and e, the x axis is displayed in wave number (nm^{-1}), while in the supplementary materials similar figures (S4 b, c, e and f) are displayed in ("spatial"?) frequency but the unit is (nm^{-1}). More consistency in the edition would make the reading easier. Consider adding the word "spatial" in front of frequency.

The Reviewer was very attentive for noticing this discrepancy of axes labeling. We decided to also label the x-axis of the Fourier transforms in the Supplementary Information with "Wave number / nm^{-1} " according to the Fourier transformations presented in the main text:

Supplementary Figure 5 | Surface morphology and fast Fourier transformation. The parallel-scanned first-order DFB resonators were fabricated with pulse distances of a) - c) 183 nm, and d) - f) 193 nm. Two analysis lines were selected from each sample and checked the amplitude and phase difference to evaluate errors. b) and e) give the dominant periodicity of 182 nm and 192 nm, and are identical between line 1 and 2, as well as line 3 and 4. c) and f) indicate the phase-locking between different raster scans with a phase difference of only 8° and 2° for the two samples. These supplemented data support our statement in the main text of high fabricating accuracy and reproducibility for every individual scan.

6. Check possible inconsistencies also in Fig 1.e and Fig 4.c and f.

We thank the Reviewer for also pointing out this inconsistency on the y-axis. We decided to change the y-axis in the Supplementary Information from degree to radiant:

Supplementary Figure 5 | Surface morphology and fast Fourier transformation. The parallel-scanned first-order DFB resonators were fabricated with pulse distances of a) - c) 183 nm, and d) - f) 193 nm. Two analysis lines were selected from each sample and checked the amplitude and phase difference to evaluate errors. b) and e) give the dominant periodicity of 182 nm and 192 nm, and are identical between line 1 and 2, as well as line 3 and 4. c) and f) indicate the phase-locking between different raster scans with a phase difference of only 8° and 2° for the two samples. These supplemented data support our statement in the main text of high fabricating accuracy and reproducibility for every individual scan.

7. Figure 3.a and the inset show emitted spectra with increasing pumping intensity. However it is very hard for the reader to really distinguish or even get an approximation of the FWHM, nor is it possible to figure out for which pump intensity it has been obtained. A larger figure and inset would improve the paper.

We thank the reviewer for pointing out these aspects of improvement. We increased the size of the inset to allow for a better FWHM understanding and added a color bar for the corresponding pumping intensities:

Fig. 1 | Fabrication of LIPSS. **a** Schematic of femtosecond laser scanning to produce laser-induced periodic surface structures (LIPSS), where the direction of the resulting LIPSS grating lines is perpendicular to the polarization of the incident laser beam. **b** Dependency of resulting LIPSS periodicity on the lateral structuring laser pulse distance. The resulting LIPSS periodicity is larger than the laser pulse spacing in regime A. In regime B, the periodicity equals the pulse spacing, whereas in regime C, no grating can be created. Regime D denotes gratings produced

by ablation (writing grating lines directly with the laser). **c** Exemplary top view SEM image with the green line marking the pixels analyzed for periodicity and phase of the LIPSS gratings. **d** Amplitudes of the Fourier spectrum of 13 analyzed LIPSS gratings with a lateral distance of 1.47 μm and **e** the corresponding Fourier phases. See Supplementary Figure 5 and 8 for more details.

8. Fig 5. a shows the amplitude of the corrugations which is a key parameter of the DFB cavities, but it is partly hidden by the orange curve. Improving the visibility of the dark curve (Realistic LIPSS grating profile) would help the readers to access the non-uniformity and the amplitude of the corrugation.

The Reviewer made a very good suggestion to significantly improve the appearance of the figure. We moved the curve of the realistic grating profile in front of the sinusoidal grating and additionally increased the thickness to make it easier for the reader to get an understanding of the non-uniformity:

Fig. 5 | Simulations to estimate the effective mode volume of lasing by comparing reflectivities of perfect and realistic gratings. **a** Experimentally measured LIPSS profile and equivalent ideal sinusoidal grating profile. **b** Simulated spectral peak reflectivities for varying grating lengths. The reflectivity of the realistic profile saturates at a significantly lower value than the sinusoidal grating. Comparing the saturation value with the reflectivity of the sinusoidal grating leads to the conclusion that the LIPSS grating can be effectively described as a sinusoidal grating with a length of 28 μm . Contour plots of reflectivity spectra as a function of grating length for **c** ideal and **d** realistic LIPSS structures.

9. Fig 5.b, c and d: x axis is displayed in grating length / period. This might be confusing? There is possible confusion: (Length of the cavity in μm divided by the period in μm → number of periods?) or (Grating length expressed in terms of period?). This casts doubt on other figures

which doesn't give justice to the experimental results. In fig 5.a the x-axis is $X/\mu\text{m}$ whereas it is in apparently in μm ($4\ \mu\text{m} / 21$ period gives $190\ \text{nm}$ which is indeed approximately the period of the grating). Consider using a different symbol than "/" to express units...; [μm] or for example (height in nm) (Grating length in period) or simply (number of periods). For other figures, the risk of confusion is less, but consistencies among the figures is important.

The Reviewer pointed out a very important point of possible data misinterpretation by the reader which we were not aware of. "period" was indeed meant to serve as the unit, so the x-axis shows grating length expressed in terms of periods. We decided to label the x-axis "Number of periods" which should make the message clearer now:

Fig. 5 | Simulations to estimate the effective mode volume of lasing by comparing reflectivities of perfect and realistic gratings. **a** Experimentally measured LIPSS profile and equivalent ideal sinusoidal grating profile. **b** Simulated spectral peak reflectivities for varying grating lengths. The reflectivity of the realistic profile saturates at a significantly lower value than the sinusoidal grating. Comparing the saturation value with the reflectivity of the sinusoidal grating leads to the conclusion that the LIPSS grating can be effectively described as a sinusoidal grating with a length of $28\ \mu\text{m}$. Contour plots of reflectivity spectra as a function of grating length for **c** ideal and **d** realistic LIPSS structures.

C. Some key data or information are missing in the main paper although some of them are presents in the supplementary materials)

10. The amplitude of the ripples is not explicitly indicated apart from the supplementary material (Table S1), whereas it is key in cavity design. This data is really missing to the reader in the first part of the document where the pattern is described. It can also be added on figure 2.c

where a vertical scale would be useful. Also, explicit indications of both the amplitude of the ripples to be compared to the thickness of the organic layer (indicated line 172).

We thank the Reviewer for noticing the missing information about the amplitude of the LIPSS ripples. We conducted multiple AFM scans on LIPSS structures fabricated with varying pulse fluence as presented in Rebuttal Figure 6. The resulting depths of the surface corrugations are in the range of 30 to 40 nm. This change is not very significant since the parameter range for successful LIPSS fabrication is very limited.

To address the Reviewer's notice about the missing information in the main document, we added this sentence at the beginning of the results part:

The depth of the surface ripples increases with the laser fluence from around 30 up to 40 nm (see Supplementary Figure 3), and the width of the LIPSS stripe varies from 130 to 400 nm (see Supplementary Figures 2 and 9).

Rebuttal Figure 6 | LIPSS grating amplitude. AFM scans of 4 samples prepared with the same shot-to-shot spacing of 189 nm but with varying pulse fluence. The lines in each image mark where the height profiles shown below were taken from. Analysis of the height profile measurements reveals average depths of 32.5 nm for 70 nJ pulse fluence, 34.8 nm for 80 nJ, 36.0 nm for 90 nJ, and 38.6 nm for 100 nJ.

To the Supplementary Information we added this figure containing the extracted height profiles for different laser fluences:

Supplementary Figure 3 | LIPSS grating amplitude. 4 samples were prepared with the same shot-to-shot spacing of 189 nm but with varying pulse fluence. The presented AFM height profile measurements reveal average depths of 32.5 nm for 70 nJ pulse fluence, 34.8 nm for 80 nJ, 36.0 nm for 90 nJ, and 38.6 nm for 100 nJ.

The corrugation depth in the presented cross-section in Figure 2c is around 33 nm. To highlight the relation between corrugation depth and thickness of the organic layer, we added both values directly into the image:

Fig. 2 | Deposition of organic gain material. **a** Chemical structures of Alq₃ and DCM. **b** Schematic representation of sample fabrication: LIPSS fabrication with focused pulsed laser beam followed by thermal evaporation of 500 nm Alq₃:DCM on top of the structures. **c** Cross-sectional SEM image of a LIPSS grating with the deposited organic layer on top. The gold and platinum layers are deposited as conductive and protection layers for the cutting process and SEM observation.

11. The width of the stripes is not indicated. It is useful for the calculation of the mode volume.

The Reviewer is right about the relevance of the LIPSS stripe width for the lasing mode formation and hence for the mode volume. However, our simulations are conducted in two dimensions only, since we are interested in the effect of the stripe length on the

reflectivity and consequently the resulting lasing mode volume. Additionally, conducting such simulation in 3 dimensions is significantly more complicated and more computing power is needed.

The fabrication parameter range to produce LIPSS is relatively small and therefore, the width of the LIPSS stripe can be varied only by a factor of around 2. See Rebuttal Figure 7 for an example:

Rebuttal Figure 7 | LIPSS stripe width variation. Both LIPSS gratings were fabricated with the same periodicity of 189 nm but with different laser pulse fluence (76 nJ for a) and 90 nJ for b)). The width varies from 320 nm for a) to 405 nm for b).

Since the width of the LIPSS gratings not discussed in the submitted draft, we decided to supply more information on the relevance of the LIPSS stripe width on the lasing behavior. A larger stripe width simply increase the total area which serves as a Bragg reflector for the lasing wavelength. This should directly influence the lasing threshold. However, detailed comparisons reveal that additional effects play a role: wider LIPSS structures are achieved by applying higher pulse fluence during the fabrication process, which quickly results in irregularities of the LIPSS, especially for both very low and very high intensities. In Rebuttal Figure 8 we show such a comparison where it can be seen that the lasing threshold indeed decreases as the LIPSS structure gets wider as long as structure still appears very regular (subfigures a) and b)). But when the LIPSS structure has visible irregularities, the observed lasing threshold is higher compared to a smaller width with better regularity (subfigures c) and d)). The irregularities in Rebuttal Figure 8d result in gaps or local phase jumps of the Bragg grating, which lowers the reflectivity and also influences the formation of the lasing mode volume. Additionally, the curvature of the grating rims limits the feedback since there will be a phase shift between the center and edge of the LIPSS grating.

To eliminate the influence of the LIPSS width as much as possible, we produced multiple samples of the same structure but with different structuring laser fluence. By visually comparing the SEM images, we chose the structures which are most similar regarding stripe width and regularity to be compared in lasing behavior.

Rebuttal Figure 8 | Comparison of different lasing thresholds for different LIPSS stripe widths. Subfigure a) and b) show 6 μm long LIPSS stripes where the stripes in b) are significantly wider. The LIPSS lines in c) and d) have a length of 17 μm . The lateral spacing between the LIPSS lines is 1.416 μm for all samples.

We mention the range of the LIPSS width in the beginning of the result chapter in the main text:

The depth of the surface ripples increases with the laser fluence from around 30 up to 40 nm (see Supplementary Figure 3), and the width of the LIPSS stripe varies from 130 to 400 nm (see Supplementary Figures 2 and 9).

And include this Figure in the Supplementary Information:

Supplementary Figure 9: Comparison of different lasing thresholds for different LIPSS stripe widths. Subfigure a) and b) show 6 μm long LIPSS stripes where the stripes in b) are significantly wider. Consequently, the measured lasing threshold is lower due to the larger volume which acts as a DFB reflector. The LIPSS lines in c) and d) have a length of 17 μm. Due to the curved appearance of the corrugation and the visible irregularities, the observed threshold is higher for d) although a larger area could serve as a DFB reflector. With increased fabrication power, the gratings are expected to become deeper and therefore enhance optical feedback. However, this is only true for a certain range, because the waveguiding efficiency is negatively influenced by high grating amplitudes¹⁰. The lateral spacing between the LIPSS lines is 1.416 μm for all samples.

Additionally, we include now the stripe widths for the presented LIPSS gratings with varied laser fabrication fluence in Supplementary Figure 2 in the Supplementary Material:

Supplementary Figure 2 | LIPSS formation under different structuring laser fluence. All samples were fabricated with the same shot-to-shot spacing of 194 nm and the same speed of 1 mm/s while the fluence of the impinging laser beam was varied (a: 70 nJ, b: 86 nJ, c: 104 nJ). Fourier analysis of the SEM-scans reveals that the dominant periodicities are a) 193.90 nm, b) 193.11 nm, and c) 194.87 nm. The resulting widths of the LIPSS stripes are 287 nm for a, 478 nm for b, and 489 nm for c.

12a. Line 218-222: the discussion is about the wave-guided mode. In waveguided mode the effective index of refraction is indeed lower than the refractive index so the reader is not surprised by the findings

We thank the reviewer for pointing out the non-novelty of this detail. We updated the sentence as follows:

By comparing the LIPSS periodicity Λ with the measured lasing wavelength based on the Bragg condition, we obtain an effective refractive index of $n_{\text{eff}} = 1.64$ at 618 nm. This fits well to the calculated value for the waveguided mode (see Supplementary Figure 10).

12b. A plot of the electric field distribution of the different wave-guided modes inside a 500 nm thick waveguide (without grating) would help the reader to better visualize the configuration and understand the discrepancy between the different refractive indices. Adding in the Supplementary Materials, a figure relevant for the calculation of the volume of mode mentioned later in the paper. For example, <https://www.sio.eu/oms.html> or other simulation software or applet.

We thank the Reviewer for this helpful comment! Using the proposed online solver, we could easily show that our measured effective refractive index of $n_{\text{eff}} = 1.64$ matches to the calculated TE0 mode:

Rebuttal Figure 9 | Mode solver results. a) shows the input parameters and the resulting effective refractive indices for each allowed mode, and b) shows the electric field distributions for both transverse modes.

Since this plot is very helpful for understanding the guided wave behavior, we added it to the Supplementary Information:

Supplementary Figure 10: Electric field distribution for the waveguided modes with a wavelength of 618 nm, where the resulting effective refractive index of the TE0 mode perfectly matches to the experimentally observed effective refractive index of $n_{\text{eff}} = 1.64$. It can be clearly seen that the evanescent part of the waveguided TE0 mode reaches into the surrounding layers, with a higher field strength in the glass substrate. The assumed refractive indices are $n = 1.4$ for the glass substrate¹¹ (left side), $n = 1.7$ for the Alq₃:DCM¹¹ gain material (middle), and $n = 1$ for the air (right side). These results were obtained using the online 1D mode solver OMS (<https://www.sio.eu/oms.html>). With the field strength being higher on the glass/organic interface than on the organic/air interface, the actual LIPSS corrugation is expected to supply the optical feedback for lasing (rather than the corrugation on the top surface of the organic layer).

13a. What are the hot quality factors of the different cavities? Is it possible to estimate the cold quality factors from the simulation?

We thank the Reviewer for the question. As we mention in more detail in our response to reviewer's question #13b (see below), we are unable to reliably measure the spectral linewidth above the lasing threshold, as we fall below the resolution limit of our detection system. While our finite-difference time-domain (FDTD) simulations effectively reveal the opening of the stopbands as a function of cavity length (see Figures 5c and 5d), they do not show the presence of Bragg resonances or pronounced Bragg dips in the reflection or transmission spectra for our structure designs with realistic parameter sets. The situation is further complicated by the onset of imperfections in the LIPSS corrugation profile, which deviates significantly from a harmonic shape due to the emergence of higher spatial Fourier harmonics resulting from the nonlinear nature of LIPSS formation. On the other hand, although the presence of DFB resonances cannot be measured experimentally or simulated by FDTD in our systems, it is indirectly supported by the achieved lasing, which depends on the periodicity of the LIPSS and confirms its inherent nature. This point is one of the most important messages of our paper, and we recognize that we may not have communicated it clearly enough, as reflected in the reviewer's

questions. Consequently, we have added additional statements in the manuscript to clarify this message for the reader:

We added this sentence to the discussion about the influence of LIPSS-inhomogeneities to the resonator effectiveness:

As a result, no DFB resonance can be observed in the linear regime below the lasing threshold.

And this sentence to the discussion of the simulations:

Similar to the photoluminescence spectra below the lasing threshold, no resonances can be observed in the reflectance spectrum.

In the conclusion part we highlight the message:

Although resonances in the linear regime are neither observed experimentally nor in simulations, the periodic surface height variation provides sufficient optical feedback along the LIPSS grating to reach lasing threshold in optically pumped organic DFB structures.

13b. In the current form of figure 3, the reader cannot estimate Q factors from the FWHM in Fig 3.b. because the right y-axis lacks of accuracy close to zero. A log scale would help. An indication of the Q factor somewhere in the text would help the reader. How the Q factor changes when the length of the cavity is changed.

We thank the Reviewer for the suggestion. Figure 3b illustrates the dynamics of peak intensity and FWHM as a function of pump energy across a range of intensities, covering the transition from spontaneous emission to lasing. In particular, the FWHM shows a very sharp decrease in the spectral width of the characteristic emission lines, which is clearly observed. We note, however, that above the lasing threshold, the spectral linewidth of the laser lines approaches the resolution limit of our detection system (approximately 0.2 nm).

While the spectral line shape of the laser emission, as depicted in Figures 3a, 3c, and 3d, may appear reliably resolvable, the additional spectral linewidth accumulated is primarily due to the angular integration of spatially divergent light emission from our sample which is specific to our detection system. Consequently, we prefer to maintain a linear plot of the FWHM as a function of pump energy, given the lack of spectral resolution above the lasing threshold. We have added the following text in the main text to address, why the Q-factor is not discussed in the paper:

Owing to the spectral resolution limit of our detection system (about 0.2 nm) and the large angular integration used during measurement, the recorded emission peaks are artificially broadened. As a result, this does not allow us to reliably estimate the spectral width of the lasing line (see Figure 3b).

As mentioned in our response to reviewer's question #13a (see above), our FDTD simulations do not show DFB resonances due to substantial deviations in LIPSS

corrugation from a harmonic spatial profile, as well as the onset of profile irregularities. However, the quality factor as a function of cavity length can be estimated indirectly via the threshold of lasing characteristics. This work is currently in progress and will be published elsewhere in the future.

D. Questions related to the scientific content

14. How does the corrugation amplitude as small as ~30 nm while the organic layer thickness is 500 nm, affect the reflectivity and the lasing action?

We thank the Reviewer for this question. The quality of the substrate corrugation profile in DFB lasers, including its depth, plays a critical role in determining the emission and lasing characteristics. Specifically, the depth of the corrugation affects the amount of optical feedback, which directly impacts the overall performance and emission properties of the laser. The effect of corrugation depth on lasing characteristics has been studied in previous works (e.g., Tsutsumi *et al.*¹⁰)

Higher corrugation amplitudes generally lead to increased diffraction efficiency, which enhances positive optical feedback from the resonator to the gain medium. However, at a certain point, excessively high corrugation amplitudes begin to substantially degrade waveguide performance, ultimately deteriorating the lasing characteristics of the laser. This is particularly pronounced in our LIPSS systems, as the corrugation profiles are far from ideal and introduce significant parasitic scattering, which acts as a loss mechanism. There is a complex interplay between the substrate corrugation depth, the waveguide, and the properties of the gain medium. While the corrugation amplitude is undoubtedly an important parameter and should ideally be optimized experimentally by comparing samples with varying corrugation depths, this is challenging for LIPSS lasers. Unfortunately, we have limited control over the substrate corrugation amplitude, as the formation of LIPSS requires strict micromachining conditions that cannot be easily varied. We believe that in our case, lasing is achieved primarily due to the excellent optical properties of our Alq₃:DCM organic gain medium and the miniaturized geometry of the lasing structures.

Fabricating deeper gratings requires the use of higher pulse fluence, where the width of the grating stripe is also increased at the same time. For very high fabrication pulse fluences, we notice an increase in lasing threshold, which is explained by the irregularities of the structures. Additionally, the lowered waveguiding efficiency due to the high grating amplitude might also play a role here.

We include this Figure in the Supplementary Information, which also addresses question 11:

Supplementary Figure 9: Comparison of different lasing thresholds for different LIPSS stripe widths. Subfigure a) and b) show 6 μm long LIPSS stripes where the stripes in b) are significantly wider. Consequently, the measured lasing threshold is lower due to the larger volume which acts as a DFB reflector. The LIPSS lines in c) and d) have a length of 17 μm . Due to the curved appearance of the corrugation and the visible irregularities, the observed threshold is higher for d) although a larger area could serve as a DFB reflector. With increased fabrication power, the gratings are expected to become deeper and therefore enhance optical feedback. However, this is only true for a certain range, because the waveguiding efficiency is negatively influenced by high grating amplitudes¹⁰. The lateral spacing between the LIPSS lines is 1.416 μm for all samples.

1. Does the upper corrugation at the organic/air interface play a role in the laser threshold?

The Reviewer asks a very good question. Even though the refractive index contrast at the top interface (organic/air) is higher than on the bottom interface (glass/organic) and the corrugations appear to be smoother in the cross-section SEM, the analysis of the effective refractive index suggests that the actual feedback is provided by the bottom interface since the measured value of $n_{\text{eff}} = 1.64$ lies between the values for glass ($n = 1.4^{11}$) and the organic ($n = 1.7^{11}$). Additionally, the field distribution in Supplementary Figure 10 shows a significantly higher extension of the electric field into the glass substrate, which suggests stronger coupling between the standing wave and the LIPSS corrugation on the organic/glass interface compared to the organic/air interface.

An influence of the upper corrugation on the lasing threshold could be that it slightly increases the threshold since the induced reflection has a phase shift of roughly π to the bottom corrugation. Also, the corrugation could slightly lower the waveguiding efficiency of the layer, but that is rather unlikely since the bottom corrugation has significantly more irregularities, hence it is the limiting factor regarding waveguiding.

We added the relevant points about this discussion to the caption of Supplementary Figure 10:

Supplementary Figure 10: Electric field distribution for the waveguided modes with a wavelength of 618 nm, where the resulting effective refractive index of the TE0 mode perfectly matches to the experimentally observed effective refractive index of $n_{\text{eff}} = 1.64$. It can be clearly seen that the evanescent part of the waveguided TE0 mode reaches into the surrounding layers, with a higher field strength in the glass substrate. The assumed refractive indices are $n = 1.4$ for the glass substrate¹¹ (left side), $n = 1.7$ for the Alq₃:DCM¹¹ gain material (middle), and $n = 1$ for the air (right side). These results were obtained using the online 1D mode solver OMS (<https://www.sio.eu/oms.html>). With the field strength being higher on the glass/organic interface than on the organic/air interface, the actual LIPSS corrugation is expected to supply the optical feedback for lasing (rather than the corrugation on the top surface of the organic layer).

2. Does the phase difference of roughly π between the bottom and top layer modification play a role in the threshold and in the quality factor?

We thank the Reviewer for the question. We do not expect the top corrugation to play a significant role at all on the optical feedback to start lasing. However, since it is a corrugation with the exact same main periodicity as the LIPSS grating, it cannot be ignored. Since the electric field component of the waveguided TE0 mode (see Rebuttal Figure 9) is reaching further into the glass substrate than into the air and the field strength is stronger at the surface with the LIPSS than at the organic/air interface, we expect the LIPSS in the glass/organic interface to be the corrugation that supplies the relevant feedback to the waveguided mode.

This leads to the result, that the top corrugation likely has no significant effect or even slightly lowers the reflection efficiency due to the phase shift of roughly π . In that case, a wave reflected on the top corrugation would cancel out with a wave reflected on the bottom corrugation. But we observe good lasing behavior which leads to the conclusion

that the effect of the face shift between top and bottom corrugation is minor on both threshold and quality factor.

15. Is it possible to enhance the ripple amplitude? Would it help to improve the reflectivity or the quality factor?

We thank the Reviewer for this question. The ripple amplitude can be controlled by varying the laser fluence of the structuring laser during LIPSS fabrication. An increase of LIPSS depth with higher laser fluence is observed as shown in Rebuttal Figure 10. Since the parameter range to fabricate LIPSS is very small, the ripple amplitude can be varied only by a small range (from roughly 30 nm to 40 nm).

Rebuttal Figure 10 | AFM pictures and the surface profiles of LIPSS structures fabricated with different laser pulse fluences. (a) 70 nJ, (b) 80 nJ, (c) 90 nJ, and (d) 100 nJ. The average depths extracted from the surface profiles are 32.5 nm, 34.8 nm, 46.0 nm, and 38.6 nm, respectively.

Since the comparison of AFM images presented in Rebuttal Figure 10 might be helpful for the reader to better understand the grating depth variations on LIPSS, we decided to add the Figure to the Supplementary Information:

Supplementary Figure 3 | LIPSS grating amplitude. 4 samples were prepared with the same shot-to-shot spacing of 189 nm but with varying pulse fluence. The presented AFM height profile measurements reveal average depths of 32.5 nm for 70 nJ pulse fluence, 34.8 nm for 80 nJ, 36.0 nm for 90 nJ, and 38.6 nm for 100 nJ.

The quality of the substrate corrugation profile in DFB lasers, including its depth, plays a critical role in determining the emission and lasing characteristics. Specifically, the depth of the corrugation affects the amount of optical feedback, which directly impacts the overall performance and emission properties of the laser. The effect of corrugation depth on lasing characteristics has been studied in previous works (e.g., Tsutsumi *et al.*¹⁰)

Higher corrugation amplitudes generally lead to increased diffraction efficiency, which enhances positive optical feedback from the resonator to the gain medium. However, at a certain point, excessively high corrugation amplitudes begin to substantially degrade waveguide performance, ultimately deteriorating the lasing characteristics of the laser. This is particularly pronounced in our LIPSS systems, as the corrugation profiles are far from ideal and introduce significant parasitic scattering, which acts as a loss mechanism. There is a complex interplay between the substrate corrugation depth, the waveguide, and the properties of the gain medium. While the corrugation amplitude is undoubtedly an important parameter and should ideally be optimized experimentally by comparing samples with varying corrugation depths, this is challenging for LIPSS lasers. Unfortunately, we have limited control over the substrate corrugation amplitude, as the formation of LIPSS requires strict micromachining conditions that cannot be easily varied. We believe that in our case, lasing is achieved primarily due to the excellent optical properties of our Alq₃:DCM organic gain medium and the miniaturized geometry of the lasing structures.

Fabricating deeper gratings requires the use of higher pulse fluence, where the width of the grating stripe is also increased at the same time. For very high fabrication pulse fluences, we notice an increase in lasing threshold, which is explained by the irregularities of the structures. Additionally, the lowered waveguiding efficiency due to the high grating amplitude might also play a role here.

We include this Figure in the Supplementary Information, which also addresses question 11:

Supplementary Figure 9: Comparison of different lasing thresholds for different LIPSS stripe widths. Subfigure a) and b) show 6 μm long LIPSS stripes where the stripes in b) are significantly wider. Consequently, the measured lasing threshold is lower due to the larger volume which acts as a DFB reflector. The LIPSS lines in c) and d) have a length of 17 μm . Due to the curved appearance of the corrugation and the visible irregularities, the observed threshold is higher for d) although a larger area could serve as a DFB reflector. With increased fabrication power, the gratings are expected to become deeper and therefore enhance optical feedback. However, this is only true for a certain range, because the waveguiding efficiency is negatively influenced by high grating amplitudes¹⁰. The lateral spacing between the LIPSS lines is 1.416 μm for all samples.

16. Line 120 indicates "we did not notice a dependency of Λ on the laser fluence (see Supplementary Figure 2)". Does the amplitude of the corrugation depend on the laser fluence?

We thank the Reviewer for this question. The ripple amplitude indeed depends on the laser fluence of the structuring laser during LIPSS fabrication. An increase of LIPSS depth with higher laser fluence is observed as shown in Rebuttal Figure 11. Since the parameter range to fabricate LIPSS is very small, the ripple amplitude can be varied only by a small range (from roughly 30 nm to 40 nm).

Rebuttal Figure 11 | AFM pictures and the surface profiles of LIPSS structures fabricated with different laser pulse fluences. (a) 70 nJ, (b) 80 nJ, (c) 90 nJ, and (d) 100 nJ. The average depths extracted from the surface profiles are 32.5 nm, 34.8 nm, 36.0 nm, and 38.6 nm, respectively.

17. Does the Fourier analysis indicated line 29-30 of Fig S2 show changes in amplitude when the fluence is changed from 126 nJ, to 135 nJ, and 144 nJ)? (Amplitudes of the FFT plots) as an indication of the change in amplitude of the ripples.

We thank the Reviewer for the question. Fourier Transformations of the LIPSS lines imaged by SEM and presented in Supplementary Figure 2 are shown in Supplementary Figure 8. It can be clearly seen that the peak height in the Fourier spectrum is slightly increasing from the lowest power to the medium power. The peak intensity for the LIPSS fabricated with the highest structuring power is significantly higher, which can be explained by the visibly brighter rims and darker valleys between the rims. A second peak with the doubled wave number is notable, which is a higher harmonic originating either from the non-sinusoidal shape of the LIPSS or results from the high contrast of the image (bright regions are overexposed and dark regions are completely black – both extreme regions lack any detail). Generally, using the intensity values (number of measured reflected electrons) from an SEM image to conclude the amplitude of the LIPSS ripples is not recommended. The reflectivity during the SEM scan does not linearly depend on the depth of the structure but rather depends on the precise conditions on how the image was taken (material, electron beam energy, detector type and location, field concentration on edges, image contrast and brightness). However, the ripples are very clearly visible, which allows for very precise analysis of the periodicity. For analysis of the LIPSS depth, AFM scans like presented in Supplementary Figures 6 and 11 are more reliable.

Rebuttal Figure 12 | Fast-Fourier Transformation of LIPSS fabricated with different laser pulse fluence. Subfigures a-c) show the LIPSS structures shown in Supplementary Figure 2 in more detail (LIPSS gratings with the same pulse-to-pulse spacing but varied laser pulse fluence). The red lines mark the horizontal pixel line of the image that was used for the Fourier Transformation. The resulting frequency intensities of the Fourier Transformation of the red lines are presented in d-f). The carrier periodicities (e.g. frequencies with the highest intensity) are 193.90 nm for d), 193.11 nm for d), and 194.87 nm for f).

18. Line 138 indicates a FFT was performed on the top-view of the SEM image. Could you indicate the way it has been performed? Is the FFT performed on the entire image or on stripes centered on a particular grating, or even a line in the center of a particular grating?

We thank the Reviewer to point out that this procedure is not detailed described in the current manuscript. The scans were performed on single lines of the image (exemplary

marked in Figure 1c). The pixel lines were manually selected and then FFT was performed on the pixel intensity values of each selected line. The resulting amplitudes and phases shown in Figures 1d and e result from 13 such scans on the full-sized image of the SEM-scan shown in Figure 1c.

In Rebuttal Figure 12 we present more examples of Fourier Analysis conducted on SEM images of LIPSS structures. Additionally, we decided to add the following figure to the Supplementary Information to supply more information to the reader:

Supplementary Figure 8 | Fourier Analysis of different LIPSS gratings. The LIPSS structures were fabricated with different laser pulse fluence but the same pulse-to-pulse spacing of 194 nm. Subfigures a-c) show the LIPSS structures shown in Supplementary Figure 2 in more detail (LIPSS gratings with the same pulse-to-pulse spacing but varied laser pulse fluence). The red lines mark the horizontal pixel line of the image that was used for the Fourier Transformation. The resulting frequency intensities of the Fourier Transformation of the red lines are presented in d-f). The carrier periodicities (e.g. frequencies with the highest intensity) are 193.90 nm for d), 193.11 nm for d), and 194.87 nm for f).

19. Line 145-14: Does a 1D FFT calculated on a band (not a single line) of the image centered on a grid width equal to the grid width provide useful information about the non-uniformity of ripples, including the difference between the center and edges of the network?

We thank the Reviewer for this question. The approach of computing FFT of multiple LIPSS lines from one single image is exactly what was conducted by us to provide the Figures 1d and e. In Rebuttal Figure 13, we present the image which was used to calculate the Fourier Transformation intensities and phases presented in Figure 1d and e.

For each of the 13 LIPSS lines, a horizontal pixel line of the image was manually chosen, such that it is centered in the grating. The brightness values of each pixel line were then analyzed by Fast Fourier Transformation, which gives amplitude and phase values over the frequency spectrum. The resulting amplitudes and phases are presented in Figure 1d and e, where it can be clearly seen that all gratings show the exact same peak frequency, and the phase relation around that main frequency is stable for all gratings.

This analysis suggests, that there are no notable frequency or phase differences between the center of a grid and the edge of a grid.

Rebuttal Figure 13 | Top-view SEM-scan of a set of LIPSS-gratings which were fabricated one after another with the same parameters. For the Fourier Transformation, a horizontal pixel line in the center of each LIPSS grating was chosen and then analyzed (see Figure 1d and e). The large damages in the LIPSS gratings are spots where two femtosecond-pulses instead of one reached the substrate due to randomly appearing synchronization issues between the shutter and the structuring laser.

We added this Figure to the Supplementary Information:

Supplementary Figure 12: Top-view SEM-scan of a set of LIPSS-gratings which were fabricated one after another with the same parameters. For the Fourier Transformation, a horizontal pixel line in the center of each LIPSS grating was chosen and then analyzed (see Figure 1d and e). The large damages in the LIPSS gratings are spots where two femtosecond-pulses instead of one reached the substrate due to randomly appearing synchronization issues between the shutter and the structuring laser.

20. Line 255-256: "Surface profile disturbance affects the optical quality due to spectral noise onset": How the lateral variation of the stripe induce noise? Would a 2D FF analysis help to answer this issue?

We thank the Reviewer for this question. Every feature in the real-space profile, which is a deviation from a perfectly sinusoidal function, can only be described by multiple additional frequencies in the Fourier spectrum. On one hand, we have a lot of point-defects in our gratings, which are mostly appearing as noise over a large frequency range in the Fourier spectrum. On the other hand, a small phase jump of the main frequency somewhere along the grating will appear as rather intense frequencies that are just slightly different to the carrier frequency. This will be notable as a widening of the main peak. Even though our LIPSS gratings show numerous visual imperfections, the Fourier spectrum (which should somewhat better represent what waveguided electromagnetic waves would experience) we see extremely clear peaks and good phase relations.

A 2D Fourier Transformation is not expected to help in characterizing phase disturbance and point defects. The advantage of 2D FFT would be to see if there are significant resonances in other directions than along the LIPSS grating. If such resonances would appear and if they would be sufficiently strong, they could possibly also serve as lasing

resonators. However, since the LIPSS is a first-order feedback structure (lower-order feedback structures allow lower lasing thresholds than higher-order ones¹²) and any repeating feature at a different angle must be a higher-order feedback structure to supply resonance within the gain spectrum, there are no such resonances expected. The approach of conducting multiple 1D-Fourier transformations on neighboring LIPSS gratings goal-oriented for understanding frequency and phase relations.

21. Line 218-222: Regarding the wave-guided modes:

1. How many modes co-exist in a 500 nm thick organic layer?

We thank the reviewer for the question. Using the online 1D mode solver OMS (<https://www.sioo.eu/oms.html>), we can show that there are two modes allowed for waveguiding in the organic layer. Rebuttal Figure 14 shows screenshots of the input parameters and the computed results, as well as a plot of the electric field distribution:

Rebuttal Figure 14 | Mode solver results. a) shows the input parameters and the resulting effective refractive indices for each allowed mode, and b) shows the electric field distributions for both transverse modes.

Since this plot is very helpful for understanding the guided wave behavior, we added it to the Supplementary Information:

Supplementary Figure 10: Electric field distribution for the waveguided modes with a wavelength of 618 nm, where the resulting effective refractive index of the TE0 mode perfectly matches to the experimentally observed effective refractive index of $n_{\text{eff}} = 1.64$. It can be clearly seen that the evanescent part of the waveguided TE0 mode reaches into the surrounding layers, with a higher field strength in the glass substrate. The assumed refractive indices are $n = 1.4$ for the glass substrate¹¹ (left side), $n = 1.7$ for the Alq₃:DCM¹¹ gain material (middle), and $n = 1$ for the air (right side). These results were obtained using the online 1D mode solver OMS (<https://www.sio.eu/oms.html>). With the field strength being higher on the glass/organic interface than on the organic/air interface, the actual LIPSS corrugation is expected to supply the optical feedback for lasing (rather than the corrugation on the top surface of the organic layer).

2. What are the values of the effective refractive index of the different waveguided modes?

Using the online 1D mode solver OMS (<https://www.sio.eu/oms.html>), the resulting effective refractive indices are 1.638 for the TE0 mode and 1.457 for the TE1 mode. See Rebuttal Figure 15 for more details:

Rebuttal Figure 15 | Mode solver results. A screenshot obtained using the online 1D mode solver OMS (<https://www.sio.eu/oms.html>), where the input parameters resemble our sample layout as close as possible.

3. Do they fit with the calculated = 1.64?

We thank the Reviewer for this detailed question. Yes, the effective refractive index of 1.638 computed by the online 1D mode solver OMS (<https://www.sio.eu/oms.html>) fits very well to our experimentally observed effective refractive index of 1.64. This shows that the waveguided TE0 mode is responsible for the lasing behavior since it couples to the LIPSS structure.

22. Line 275: The concept of Coherence propagation is unclear.

We thank the Reviewer for this note. The phrase “coherence propagation” is used for describing the observed behavior in the FDTD simulations presented in Figure 5. We want to express here, that the length of the grating which is coherently reflecting the light (e.g. all the light reflected in that portion of the grating is in phase) is limited because of randomly appearing irregularities in the grating and the overall non-homogeneity of the LIPSS. Hence, when photoluminescence in the gain medium starts and the light is immediately reflected back, the volume which the resulting standing wave covers, will grow with more photons being emitted into the standing wave. This growth can be pictured as a coherence propagating along the grating until the gratings inhomogeneities are not allowing a further extension of the standing wave any more.

To avoid possible confusion of the reader, we changed the word “propagation” in the main text to “build-up”, since we also use this word for the same expression further in the text:

Using numerical simulations, we show in Figure 5b that in our system, the spatial profile inhomogeneities are the dominant factor limiting the coherence build-up along the LIPSS grating vector.

23. Page 10 Line 295: The legend of figure 5 appears unclear: "Simulations to estimate the effective mode volume of lasing." How do the different figure relate to the effective mode volume? How do you use the dimension of the grating, and the field distribution of the different modes (Q 21) to calculate the mode volume.

We thank the Reviewer for this question. We want to emphasize that we are only interested in the dimension of the mode volume expanding in the direction of the grating vector of the LIPSS. The simulations serve the purpose of estimating what length of a perfectly shaped sinusoidal grating would be needed for the emitted light to experience the same reflectivity as it is experiencing with the non-perfect LIPSS gratings. This is of course not a flawless approach since we only simulate in one dimension (canceling out any influences by the limited grating width) and the simulation is purely linear, but lasing is extensively non-linear.

The idea is that imperfections in the LIPSS grating, such as irregularities in the amplitude and shape, and perhaps even local phase jumps, significantly lower the reflectivity of the grating, which makes it harder for lasing to start (e.g. increases the lasing threshold). These imperfections can result in the reflected light having a slightly different phase compared to the light reflected from a neighboring section. However, during the lasing, all the light within the active mode volume is perfectly in phase. We want to find out what the length of this mode volume is by comparing the reflectivity of the LIPSS grating to a perfect grating where always all reflected light is perfectly in phase.

We improved the caption on Figure 5 to highlight the purpose of those simulations:

Fig. 5 | Simulations to estimate the effective mode volume of lasing by comparing reflectivities of perfect and realistic gratings. **a** Experimentally measured LIPSS profile and equivalent ideal sinusoidal grating profile. **b** Simulated spectral peak reflectivities for varying

grating lengths. The reflectivity of the realistic profile saturates at a significantly lower value than the sinusoidal grating. Comparing the saturation value with the reflectivity of the sinusoidal grating leads to the conclusion that the LIPSS grating can be effectively described as a sinusoidal grating with a length of 28 μm . Contour plots of reflectivity spectra as a function of grating length for **c** ideal and **d** realistic LIPSS structures.

24. Page 10 also line 297: Legend of Figure 5.b use the expression "peak reflectivity" which is unclear. Why is it peak? Does it relate to the reflectivity saturation levels line 290? Does the reflectivity saturation level have the same meaning than the peak reflectivity?

We thank the Reviewer for this clarifying question. The term "peak reflectivity" in the caption of Figure 5 means that for each simulated grating length we took the reflectivity value which was the highest from all the wavelengths that were simulated. The total simulation results are shown in Figure 5c and d where the x-axis represents the length of the grating and the y axis the wavelength of the incident light. Per each grating length value, the wavelength with the highest reflectivity value was chosen and the value is presented in Figure 5b. These peak values itself are saturating with increasing grating length as expected.

To clarify the caption a bit, we added "spectral" before "peak intensities":

Fig. 5 | Simulations to estimate the effective mode volume of lasing by comparing reflectivities of perfect and realistic gratings. **a** Experimentally measured LIPSS profile and equivalent ideal sinusoidal grating profile. **b** Simulated spectral peak reflectivities for varying grating lengths. The reflectivity of the realistic profile saturates at a significantly lower value than the sinusoidal grating. Comparing the saturation value with the reflectivity of the sinusoidal grating leads to the conclusion that the LIPSS grating can be effectively described as a sinusoidal

grating with a length of 28 μm . Contour plots of reflectivity spectra as a function of grating length for **c** ideal and **d** realistic LIPSS structures.

25. Line 295: The legend of figure 5 starts with "Simulations to estimate the effective mode volume of lasing.". How do the figures 5 provide evidence of the calculation of the mode volume. It gives indications on some of the dimensions (cavity width is absent), so how does it allow to calculate the mode volume?

We thank the Reviewer for this question. We want to emphasize that we are only interested in the dimension of the mode volume expanding in the direction of the grating vector of the LIPSS. The simulations serve the purpose of estimating what length of a perfectly shaped sinusoidal grating would be needed for the emitted light to experience the same reflectivity as it is experiencing with the non-perfect LIPSS gratings. This is of course not a flawless approach since we only simulate in one dimension (canceling out any influences by the limited grating width) and the simulation is purely linear, but lasing is extensively non-linear.

The idea is that imperfections in the LIPSS grating, such as irregularities in the amplitude and shape, and perhaps even local phase jumps, significantly lower the reflectivity of the grating, which makes it harder for lasing to start (e.g. increases the lasing threshold). These imperfections can result in the reflected light having a slightly different phase compared to the light reflected from a neighboring section. However, during the lasing, all the light within the active mode volume is perfectly in phase. We want to find out what the length of this mode volume is by comparing the reflectivity of the LIPSS grating to a perfect grating where always all reflected light is perfectly in phase.

To better highlight our intentions with this approach, we changed the following paragraph in the main text:

The ability to precisely control and easily scale the size of laser-inscribed photonic structures allows us to systematically investigate lasing characteristics as a function of resonator geometry. A larger mode volume enables more molecules to participate in the lasing process, leading to enhanced output intensity. Simultaneously, the increased net gain associated with a larger mode volume reduces the lasing threshold. Exploiting the high versatility of our fabrication method, we prepared a series of samples with varying LIPSS stripe lengths to precisely control the DFB grating length, as shown in Figure 4a. **With this approach, we are altering one dimension of the lasing mode volume and aim to understand the influence of the cavity quality on the lasing behavior.** As the stripe length decreases, the mode volume becomes constrained by the size of the inscribed feedback structure. When the stripe length falls below the effective mode volume, a pronounced increase in the lasing threshold is expected. Figure 4b shows the lasing threshold energy density as a function of the LIPSS grating length, where a clear decrease of lasing threshold for longer stripe lengths with subsequent saturation is visible.

References for the rebuttal letter:

1. Bonse, J., Krüger, J., Höhm, S. & Rosenfeld, A. Femtosecond laser-induced periodic surface structures. *Journal of Laser Applications* **24**, 042006 (2012).
2. Gräf, S., Kunz, C. & Müller, F. A. Formation and Properties of Laser-Induced Periodic Surface Structures on Different Glasses. *Materials* **10**, 933 (2017).
3. Shi, X. & Xu, X. Laser fluence dependence of ripple formation on fused silica by femtosecond laser irradiation. *Appl. Phys. A* **125**, 256 (2019).
4. Ben-Yakar, A. *et al.* Morphology of femtosecond-laser-ablated borosilicate glass surfaces. *Applied Physics Letters* **83**, 3030–3032 (2003).
5. Liang, F., Vallée, R. & Chin, S. L. Mechanism of nanograting formation on the surface of fused silica. *Opt. Express, OE* **20**, 4389–4396 (2012).
6. Sun, Q., Liang, F., Vallée, R. & Chin, S. L. Nanograting formation on the surface of silica glass by scanning focused femtosecond laser pulses. *Opt. Lett., OL* **33**, 2713–2715 (2008).
7. Richter, S., Heinrich, M., Döring, S., Tünnermann, A. & Nolte, S. Formation of femtosecond laser-induced nanogratings at high repetition rates. *Appl. Phys. A* **104**, 503–507 (2011).
8. Siegman, A. E., Sasnett, M. W. & Johnston, T. F. Choice of clip levels for beam width measurements using knife-edge techniques. *IEEE Journal of Quantum Electronics* **27**, 1098–1104 (1991).
9. Liang, F., Vallée, R., Gingras, D. & Chin, S. L. Role of ablation and incubation processes on surface nanograting formation. *Opt. Mater. Express, OME* **1**, 1244–1250 (2011).
10. Tsutsumi, N., Nagi, S., Kinashi, K. & Sakai, W. Re-evaluation of all-plastic organic dye laser with DFB structure fabricated using photoresists. *Sci Rep* **6**, 34741 (2016).
11. Kozlov, V. G., Bulović, V., Burrows, P. E. & Forrest, S. R. Laser action in organic semiconductor waveguide and double-heterostructure devices. *Nature* **389**, 362–364 (1997).
12. Li, Y. & Lakhwani, G. Distributed feedback lasers up to the 400th Bragg order with an organic active layer. *Applied Physics Letters* **122**, 021108 (2023).

Reviewer #1:

The authors responded to all my suggestions and questions in a very detailed and thorough response letter. This also applies to the other reviewers. Due to the interesting topic, I propose the revised manuscript for publication.

We thank the Reviewer for this very positive feedback and for contributing to the review of our manuscript!

Reviewer #2:

We thank the Reviewer for contributing to the review of our manuscript.

Reviewer #4:

Dear Authors,

Your detailed answers to my comments and questions about your manuscript entitled "Read my LIPSS: organic lasers on micromachined resonators" are convincing. The changes made to the manuscript greatly clarified the points I had raised and improved the quality of the article.

In particular it is now much easier for the reader to evaluate the strengths and limits of the LIPSS method for the fabrication of DFB cavity for organic lasers.

I am now convinced that your manuscript meets the criteria for publication in Nature Com and I recommend it for publication. I think your work will contribute to the field of organic lasers.

Sincerely,

We thank the Reviewer for this kind answer and thank him or her for the detailed review of our work.